# DP-Mix: Mixup-based Data Augmentation for Differentially Private Learning

**Wenxuan Bao**[1]    **Francesco Pittaluga**[2]    **Vijay Kumar B G**[2]    **Vincent Bindschaedler**[1]

[1] University of Florida    [2] NEC Labs America

## Abstract

Data augmentation techniques, such as image transformations and combinations, are highly effective at improving the generalization of computer vision models, especially when training data is limited. However, such techniques are fundamentally incompatible with differentially private learning approaches, due to the latter's built-in assumption that each training image's contribution to the learned model is bounded. In this paper, we investigate why naive applications of multi-sample data augmentation techniques, such as mixup, fail to achieve good performance and propose two novel data augmentation techniques specifically designed for the constraints of differentially private learning. Our first technique, DP-MIX$_{\text{SELF}}$, achieves SoTA classification performance across a range of datasets and settings by performing mixup on self-augmented data. Our second technique, DP-MIX$_{\text{DIFF}}$, further improves performance by incorporating synthetic data from a pre-trained diffusion model into the mixup process. We open-source the code at `https://github.com/wenxuan-Bao/DP-Mix`.

## 1  Introduction

Differential privacy (DP) [11, 12] is a well-established paradigm for protecting data privacy, which ensures that the output of computation over a dataset does not reveal sensitive information about individuals in the dataset. However, training models with DP often incur significant performance degradation, especially when strong privacy guarantees are required or the amount of training data available is limited.

One class of techniques to overcome limited training data in classical (non-private) learning is data augmentation. Unfortunately, the analysis of differentially private learning mechanisms requires that the influence of each training example be limited, so naively applying data augmentation techniques is fundamentally incompatible with existing differential private learning approaches. Therefore, new approaches are required to leverage data augmentation.

Recently, De et al. [9] showed one way to incorporate simple data augmentation techniques, such as horizontal flips and cropping, into differentially private training. Their approach creates several *self-augmentations* of a training example and averages their gradients before clipping. This procedure is compatible with differential privacy because each training example only impacts the gradients of the self-augmentations which are aggregated together before clipping, thus preserving the sensitivity bound. However, this requires using data augmentation techniques that operate on a single input example, thus excluding large classes of multi-sample techniques such as mixup [46, 16, 38, 47, 6] and cutmix [44, 8]. Furthermore, as shown in [9] their approach quickly hits diminishing returns after 8 or 16 self-augmentations.

37th Conference on Neural Information Processing Systems (NeurIPS 2023).

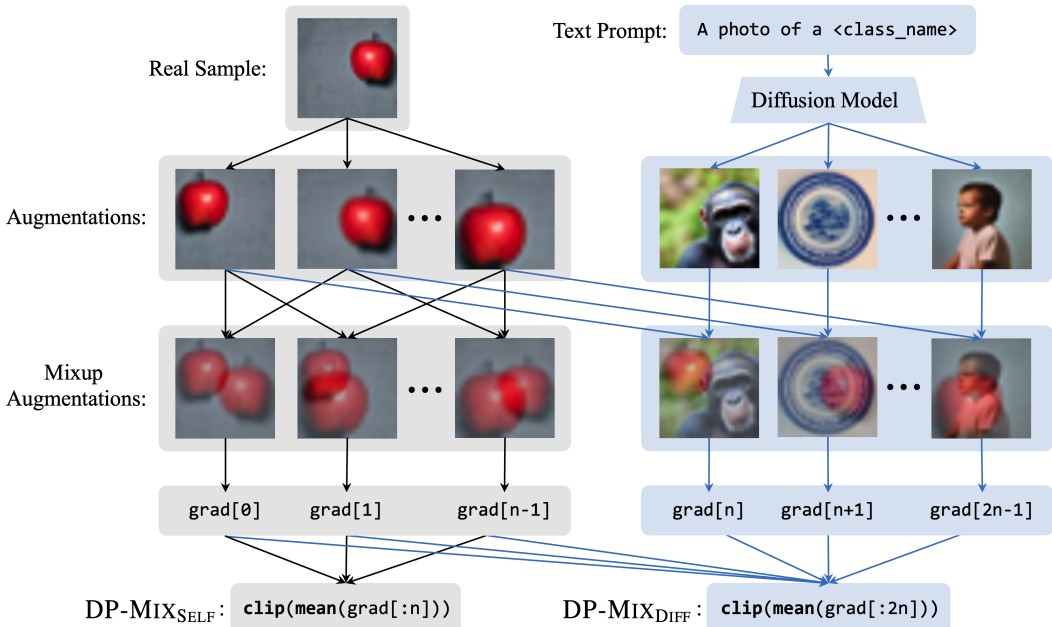

Figure 1: **Approach Overview**. Illustration of ur proposed methods for applying mixup under differential privacy: DP-MIX$_{\text{SELF}}$ and DP-MIX$_{\text{DIFF}}$.

In this paper, we leverage multi-sample data augmentation techniques, specifically mixup, to improve differentially private learning[1]. We first show empirically that the straightforward application of mixup, i.e., using microbatching, fails to improve performance, even for modest privacy budgets. This is because microbatching inherently requires a higher noise level for the same privacy guarantee.

To overcome this, we propose a technique to apply mixup in the per-example DP-SGD setting by applying mixup to self-augmentations of a training example. This technique achieves SoTA classification performance when training models from scratch and when fine-tuning pre-trained models. We also show how to further enhance classification performance, by using a text-to-image diffusion-based model to produce class-specific synthetic examples that we can then mixup with training set examples at no additional privacy cost.

## 2   Background & Related Work

### 2.1   Differential Privacy & DP-SGD

Differential privacy (DP) [11] has become the standard framework for provable privacy guarantees.

**Definition 1** (($\varepsilon, \delta$)-Differential Privacy)**.** *A mechanism $\mathcal{M}$ (randomized algorithm) satisfies $(\varepsilon, \delta)$-differential privacy if for any neighboring datasets $D$, $D'$ and any output set $S \subseteq \text{Range}(\mathcal{M})$, we have:* $\Pr(\mathcal{M}(D \in S) \leq \exp(\varepsilon) \Pr(\mathcal{M}(D' \in S) + \delta$ .

In the definition $\varepsilon > 0$ is called the privacy budget and $0 < \delta < 1$. Datasets $D, D'$ are neighboring if one can be obtained from the other by adding or removing one data point.

**DP-SGD.**   To train a machine learning model with differential privacy (Definition 1), we use Differentially Private Stochastic Gradient Descent (DP-SGD). DP-SGD was proposed by Abadi et al. [2] as a direct replacement for SGD. Informally, for differential privacy to be satisfied, each weight update must hide the presence or absence of any example in the training set (and thus the minibatch). Consider a minibatch $B$ with $n$ examples. The weight update equation of (minibatch) SGD is: $w_{j+1} = w_j - \eta g_j(B)$, where $g_j(B) = \nabla_w l(B; w_j)$ is the gradient of the loss on $B$ with

---

[1]We open-source code at `https://github.com/wenxuan-Bao/DP-Mix`.

respect to the parameters $w$ and $\eta$ is the learning rate. Remark that inclusion or exclusion of a single example in minibatch $B$ can completely change the minibatch aggregate gradient $g_j(B)$. To avoid this and satisfy differential privacy, DP-SGD first clips individual gradients and then adds Gaussian noise to the average gradient:

$$\hat{g}(B) = \frac{1}{n} \left[ \sum_{i=1}^{n} \mathrm{clip}_C\{g_i\} + \mathcal{N}(0, C^2\sigma^2) \right] , \tag{1}$$

where $g_i$ is the gradient of the loss on example $i$ of minibatch $B$ with respect to the parameters and $\mathrm{clip}_C\{g\} = g \cdot \min\{1, \frac{C}{||g||_2}\}$, where $C > 0$ is a constant. By clipping individual example gradients to norm $C$, the sensitivity of the gradient sum to adding or removing an example is guaranteed to be at most $C$. After that, adding Gaussian noise with standard deviation $C\sigma$ satisfies DP because it is an application of the Gaussian mechanism [12]. The noise level $\sigma$ is calibrated to the privacy parameters $\varepsilon$ and $\delta$.

Since the gradient update only touches the minibatch $B$, which was sampled from the entire training dataset, we can use amplification by sampling theorems [3] with a sampling rate of $n/N$ where $N$ is the size of the dataset and $n = |B|$. To obtain a differential privacy guarantee over the entire training procedure (which potentially touches each training example multiple times), we apply composition results [2]. The full procedure is presented in Algorithm 2 in the supplementary material.

**Microbatch DP-SGD.** Abadi et al. [2] proposed DP-SGD with per-example gradient clipping. But other researchers have proposed microbatch clipping instead [23], which views a minibatch $B$ as composed of $m$ microbatches, each of $n/m$ examples. With microbatch clipping, we clip the average of the microbatch's gradients instead of the per-example gradients. Note that using microbatches of size 1 is equivalent to per-example clipping.

The noisy gradient for a microbatch of size $b > 1$ (assuming $b$ divides $n$) is:

$$\hat{g} = \frac{b}{n} \left[ \sum_{i=1}^{n/b} \mathrm{clip}_C \left\{ \frac{1}{b} \sum_{j=1}^{b} g_{i,j} \right\} + \mathcal{N}(0, C^2\sigma^2) \right] , \tag{2}$$

where $g_{i,j}$ denotes the gradient vector the $j^{\mathrm{th}}$ example in the $i^{\mathrm{th}}$ microbatch.

The drawback of this approach is that despite clipping microbatches gradients with a clipping norm of $C$, the sensitivity of adding or removing one example is actually $2C$ (as pointed out recently by Ponomareva et al. [29]). On a per-example basis, more noise is added.

**Data Augmentation and DP.** Data augmentation cannot be straightforwardly applied with differential privacy because the privacy analysis requires a bound on the sensitivity of any example in the training set. For example, naively applying data augmentation prior to training, by say creating $k$ augmentations of each training set example, compromises the privacy guarantee. Existing privacy analyses do not apply because adding or removing an example in the training set now potentially affects the gradients of $k$ examples in each training epoch.

De et al. [9] proposed a way to use *single-sample* data augmentation. Their idea is to do per-example clipping but to create augmentations of each example (e.g., horizontal flip and cropping) and average the corresponding gradient before clipping. This is compatible because each training set example still only affects one term of the sum of Eq. (1). However, this idea cannot be used for more sophisticated *multi-sample* data augmentations techniques like mixup [46] that take two or more examples as input to create each augmentation.

## 2.2 Related work.

**SoTA performance in Differential Privacy Machine Learning.** Recent studies [9, 5, 34, 22, 27] achieve new SoTA performance in differentially private learning using diverse methods. De et al. [9] modified a Wide-ResNet model's training method, advocating large batches, group normalization, weight standardization, and self-augmentation, achieving SoTA performance on CIFAR-10 with DP-SGD. Sander et al. [34] introduced a technique using Total Amount of Noise (TAN) and a scaling law to reduce the computational expense of hyperparameter tuning in DP-SGD, achieving SoTA

performance on ImageNet and computational cost reductions by a factor of 128. Bu et al. [5] presented mixed ghost gradient clipping, which eliminates per-sample gradient computations, approximates non-private optimization, and achieves SoTA performance on the CIFAR-10 and CIFAR-100 datasets by fine-tuning pre-trained transformer models. Panda et al. [27] propose an accelerated fine-tuning method, which enables them to achieve SoTA performance on the CIFAR-10 and CIFAR-100 datasets.

**Mixup techniques.** Mixup was initially proposed by Zhang et al. [46]. This technique is applied to raw input data and its corresponding one-hot encoded labels. Experimental results presented in their work indicate that Mixup enhances model performance and robustness. However, Guo et al. [16] discovered that input mixup may encounter the manifold intrusion problem when a mixed example coincides with a real sample. To address this, they proposed Adaptive Mixup, which they demonstrated outperforms the original method. To mitigate the incongruity in Mixup data, Yun et al. [44] proposed Cutmix, which replaces a portion of one sample with another. Instead of applying mixup at the input level, Verma et al. [39] implemented mixup within the intermediate layers of the model. Several works have utilized Mixup in the context of Differential Privacy (DP), such as Borgnia et al. [4], which used it as a defense against backdoor attacks, and Xiao et al. [40], which used it to introduce randomness into the training process, or in other settings such as Split learning [26]. To the best of our knowledge, no previous work has employed Mixup as a data augmentation strategy to enhance model performance in differentially private learning.

**Diffusion models.** Diffusion models [36, 19, 37, 10, 24] have emerged as a powerful class of generative models that leverage the concept of stochastic diffusion processes to model complex data distributions, offering a promising alternative to traditional approaches such as Generative Adversarial Networks (GANs) [14] and Variational Autoencoders (VAEs) [20]. By iteratively simulating the gradual transformation of a simple initial distribution into the desired target distribution, diffusion models provide a principled framework for generating high-quality samples however, the implicit guidance results in a lack of user control over generated images. Recently there has been a lot of interest in text-to-image models [31, 33, 32, 13, 25, 30] due to their ability to generate entirely new scenes with previously unseen compositions solely based on natural language. In this work, we use class labels as text input to the text-to-image diffusion model [13] to generate diverse synthetic samples for augmenting the dataset.

## 3 Method

Given two examples $(x_0, y_0)$ and $(x_1, y_1)$ and mixing parameter $\lambda$, mixup creates augmentations as:

$$x = \lambda \cdot x_0 + (1 - \lambda) \cdot x_1 \quad \text{and} \quad y = \lambda \cdot y_0 + (1 - \lambda) \cdot y_1 . \tag{3}$$

In our experiments, $\lambda$ is sampled from a $\beta$ distribution with $\alpha = 0.2$. In other words, the augmented example image $x$ is simply a linear combination of two individual examples $x_0$, $x_1$ and its label $y$ is also a linear combination of the two individual examples' labels (one-hot encoded). Typically the two examples $(x_0, y_0)$ and $(x_1, y_1)$ are selected randomly from the training data. To use mixup while preserving differential privacy, we must find a way to account for its privacy impact.

For DP training, we cannot use Eq. (3) to create an augmented training set (e.g., by repeatedly taking two input examples from the sensitive training set and mixing them up) without affecting the privacy budget, as the sensitivity of the clipped-gradients sum of the augmented minibatch would then be (at least) $2C$ (and not $C$). To see why, observe that an original example $z$ would impact multiple terms of the sum in Eq. (1) (the one involving the gradient of $z$ and also the one(s) involving the gradient of any mixed-up pair(s) involving $z$). Even with exactly one mixed-up pair per original example, adding or removing an example from the training set would impact two terms in the clipped-gradients sum, potentially changing the aggregated gradient by $2C$. In a pathological case, the sum could go from $C\vec{e}$ for some $\vec{e}$ to $-C\vec{e}$. The sensitivity being $2C$ means that the scale of Gaussian noise added must be doubled to obtain the same privacy guarantee as we would have without mixup.

We empirically investigate the impact of doubling the noise for training a WRN-16-4 model on CIFAR-10. Specifically, we use a microbatch size of 2 and do input mixup by applying Eq. (3) to the two examples in each microbatch. We then add the mixed up example to the microbatch or let the microbatch consist only of the mixed up example. Then, following microbatch DP-SGD, we clip the gradient of the microbatch and add Gaussian noise to the aggregated gradient sum. Nevertheless, it

Table 1: **Test accuracy (%) of a WRN-16-4 model on CIFAR-10** trained from scratch with $(\varepsilon = 8, \delta = 10^{-5})$-differential privacy. Using a microbatch of size 2 yields much worse results than per-example clipping.

| Method | Microbatch Size = 1 | Microbatch Size = 2 |
|---|---|---|
| DP-SGD | 72.5 (.5) | 49.7 (.7) |
| DP-SGD w/ Mixup | N/A | 50.1 (.4) |
| DP-SGD w/ Self-Aug | 78.7 (.5) | 52.8 (.1) |

could be that empirically, the benefits of mixup outweigh the additional noise. Results in Table 1 show that despite mixup providing a clear benefit (compared to traditional DP-SGD) the microbatch size 2 setting is inherently disadvantaged. Better performance is obtained without mixup for per-example DP-SGD. The negative impact of noise due to the increased sensitivity dwarfs any benefits of mixup.

## 3.1 DP-Mixup

We propose a technique to apply mixup with per-example DP-SGD. The challenge is that with per-example DP-SGD we only have a single example to work with — else we again violate the sensitivity bound required for privacy analysis.

**DP-MIX$_{\text{SELF}}$ and DP-MIX$_{\text{DIFF}}$.** The basic observation behind our technique is that we can freely apply mixup to any augmentations derived from a single data point $(x, y)$ or from $(x, y)$ and synthetic samples obtained *independently* from a diffusion model. If we apply all augmentations, including mixup, *before* gradient clipping then the privacy guarantee and analysis holds according to the augmentation multiplicity insight from [9].

We assume that we have a randomized transformation function $T$ that can be applied to any example's features $x$ so that $T(x)$ is a valid single-sample data augmentation of $x$. We may also have access to a generative model from which we can produce a set $\mathcal{D}$ of synthetic samples ahead of time. For example, we can use a txt2img diffusion model to create labeled examples $(z_1, y_1), (z_2, y_2), \ldots, (z_m, y_m)$ using unguided diffusion. In experiments, we generate these samples using the text prompt 'a photo of a <class name>'. Since these "synthetic" examples are generated without accessing the (sensitive) training dataset, they can be used with no additional privacy cost.

Given a single data point $(x, y)$, our technique consists of three steps to produce a set of augmentations. The gradients of each augmentation are then computed and aggregated, and the aggregated gradient is clipped and noised as in DP-SGD. The first step takes $x$ and repeatedly applies to it the randomized transformation function $T$, resulting in a set $S = \{(x_1, y), (x_2, y), \ldots, (x_{K_{\text{BASE}}}, y)\}$ of "base" self-augmentations. The second step (optionally) augments the set $S$ with $K_{\text{DIFF}}$ randomly selected synthetic examples from the set $\mathcal{D}$ created using the diffusion model. If no diffusion model is available, we omit this step, which is equivalent to setting $K_{\text{DIFF}} = 0$. In the third and final step, we repeatedly apply Eq. (3) $K_{\text{SELF}} > 0$ times to two randomly selected samples from the augmentation set $S$, thereby obtaining a set $S'$ of $K_{\text{SELF}}$ mixed up examples. The final set of augmentations is then the concatenation of the base augmentations and the mixup augmentations, i.e., $S \cup S'$.

From this final set of augmentations, we perform DP-SGD training. That is, we compute the noisy gradient as:

$$\hat{g} = \frac{1}{n} \left[ \sum_{i=1}^{n} \text{clip}_C \left\{ \frac{1}{K} \sum_{k=1}^{K} g_{i,k} \right\} + \mathcal{N}(0, C^2 \sigma^2) \right] , \tag{4}$$

where $K = K_{\text{BASE}} + K_{\text{DIFF}} + K_{\text{SELF}}$ is the total number of augmentations and $g_{i,k}$ denotes the gradient of the $k^{\text{th}}$ augmentation of the $i^{\text{th}}$ training example. The same privacy analysis and guarantee are obtained in this case as in vanilla DP-SGD because the influence of each training example $x$ on the noisy gradient $\hat{g}$ is at most $C$ (observe that the gradients of all augmentations' are clipped together).

The method is described in Algorithm 1. An important remark is that the self-augmentation method of De et al. [9] is a special case of our method, corresponding to setting $K_{\text{DIFF}} = 0$ and $K_{\text{SELF}} = 0$, i.e., one only uses the base augmentation set and no mixup.

**Algorithm 1** DP-SGD with mixup (DP-MIX$_{\text{SELF}}$ and DP-MIX$_{\text{DIFF}}$).

---

**Require:** Training data $(x_1, y_1), ..., (x_N, y_N)$, loss function $\mathcal{L}(\theta) = \frac{1}{N} \sum_i \mathcal{L}(f_\theta(x_i), y_i)$. Parameters: learning rate $\eta_t$, noise scale $\sigma$, group size $B$, gradient norm bound $C$, number of augmentations $K_{\text{BASE}}$, number of mixup $K_{\text{SELF}}$ and number of text-to-image samples $K_{\text{DIFF}}$. Let $K = K_{\text{BASE}} + K_{\text{SELF}} + K_{\text{DIFF}}$.
    Generate txt2img samples: $\mathcal{D} = (z_1, y_1), (z_2, y_2), \ldots, (z_m, y_m)$
    **Initialize** $\theta_0$ randomly
    **for** $t \in [T]$ **do**
        Take a random sample $B_t$ with sampling probability $\frac{B}{N}$
        **for** each $(x_i, y_i) \in B_t$ **do**
            Generate S $= \{(x_{i1}, y_i), (x_{i2}, y_i), \ldots, (x_{iK_{\text{BASE}}}, y_i)\}$ // `Self-Aug`
            **if** $K_{\text{DIFF}} > 0$ **then**
                Randomly select $K_{\text{DIFF}}$ samples from $\mathcal{D}$ and add them to $S$ // `DP-Mix`$_{\text{DIFF}}$
            **end if**
            Generate $S'$ consisting of $K_{\text{SELF}}$ mixup samples from randomly selected pairs from $S$ // `DP-Mix`$_{\text{SELF}}$
            **for** each $(x^*, y^*) \in S \cup S'$ **do**
                $\mathbf{g}_t(x^*) \leftarrow \nabla_\theta \mathcal{L}(f_\theta(x^*), y^*)$
            **end for**
            $\mathbf{g}_t(x_i)' \leftarrow \frac{1}{K} \sum (\mathbf{g}_t(x^*))$
            $\bar{\mathbf{g}}_t(x_i) \leftarrow \mathbf{g}_t(x_i)' / \max(1, \frac{||\mathbf{g}_t(x_i)'||_2}{C})$
        **end for**
        $\tilde{\mathbf{g}}_t \leftarrow \frac{1}{B} \sum_i (\bar{\mathbf{g}}_t(x_i) + \mathcal{N}(0, \sigma^2 C^2 \mathbf{I}))$
        $\theta_{t+1} \leftarrow \theta_t - \eta_t \tilde{\mathbf{g}}_t$
    **end for**
**Ensure:** $\theta_T$ and compute the overall privacy cost $(\varepsilon, \delta)$ using a privacy accounting method

---

We define two variants of our method based on the availability of diffusion samples. If we have access to a diffusion model so that $K_{\text{DIFF}} > 0$, we call the method DP-MIX$_{\text{DIFF}}$. In that case, the pool from which examples are selected for mixup includes synthetic samples from the diffusion samples. Otherwise, $K_{\text{DIFF}} = 0$ and the proposed method has the effect of applying mixup on randomly selected self-augmentations of a single original example, thus we call this variant DP-MIX$_{\text{SELF}}$. Interestingly, DP-MIX$_{\text{SELF}}$ is not completely faithful to classical mixup because there is only one original example being mixed up and the label is unchanged.

Finally, we point out that although our focus is on mixup, other multi-sample data augmentation techniques such as cutmix and others [44, 8] can be applied in the third step.

## 4 Experimental Setup

**Datasets.** We use CIFAR-10, CIFAR-100, EuroSAT, Caltech 256, SUN397 and Oxford-IIIT Pet. The details of these datasets are in Appendix C in Supplemental materials.

**Models.** We use the following models/model architectures.

- **Wide ResNet (WRN)** [45] is a variant of the well-known ResNet (Residual Network) model [17], which is commonly used for image classification tasks. It increases the number of channels in convolutional layers (width) rather than the number of layers (depth). WRN-16-4 is used in De et al. [9] and we use the same model to ensure a fair comparison.
- **Vit-B-16 and ConvNext** are pre-trained on the LAION-2B dataset [35], the same pre-trained dataset as for our diffusion models, which we obtained from Open Clip[2]. We add a linear layer as a classification layer. We only fine-tune this last layer and freeze the weights of other layers.

**Setup.** To implement DP-SGD, we use Opacus [1] and make modifications to it. For training from scratch experiments, we set the batch size to 4096, the number of self-augmentation to 16, the clip bound to $C = 1$, and the number of epochs to 200. For fine-tuning experiments, we change the batch size to 1000 and the number of epochs to 20 for EuroSAT and 10 for all other datasets. Reported test accuracies are averaged based on at least three independent runs and we also report the standard deviation. We provide additional details on our experimental setups in supplementary materials.

---

[2]`https://github.com/mlfoundations/open_clip`

**Selection of pre-training data, diffusion model, and fair comparisons.** We take great care to ensure that our experiments lead to a fair comparison between our methods and alternatives such as Self-Aug (prior SoTA). In particular, all methods have access to the exact same training data. We also tune the hyperparameters of each method optimally (e.g., $K_{\text{BASE}}$ for Self-Aug). We use the same pre-trained models (Vit-B-16 and ConvNext from Open Clip) to compare our methods to others (Self-Aug and Mixed Ghost Clipping).

Since DP-MIX$_{\text{DIFF}}$ uses a diffusion model to generate synthetic examples, this could make the comparison unfair because other methods do not use synthetic samples. To avoid this, we purposefully use the exact same pre-training data (i.e., LAION-2B) to pre-train models as was used to train the diffusion model. This avoids the issue of the synthetic examples somehow "containing" data that other methods do not have access to. Moreover, we conducted experiments (check Table 6) to show that the synthetic examples themselves do *not* boost performance. It is the *way* they are used by DP-MIX$_{\text{DIFF}}$ that boosts performance. Finally, out of the six datasets we used for evaluation, none of them overlap with the LAION-2B dataset (to the best of our knowledge).

## 5 Experiments

### 5.1 Training from Scratch with DP

Since DP-MIX$_{\text{DIFF}}$ relies on synthetic samples from diffusion models, it would not be fair to compare it to prior SoTA methods that do not have access to those samples in the training from scratch setting. Therefore, we focus this evaluation on DP-MIX$_{\text{SELF}}$. For this, we use a WRN-16-4 model. Our baseline from this experiment is De et al. [9] who use a combination of techniques to improve performance, but mostly self-augmentation. Results in Table 2 show that DP-MIX$_{\text{SELF}}$ consistently provides improvements across datasets and privacy budgets.

We observed differences between the results reported by De et al. [9] using JAX and the reproduction by Sander et al. [34] on Pytorch. In our supplemental materials, Table 11 presents our reproduction of De et al. [9] using their original JAX code alongside the DP-MIX$_{\text{SELF}}$ implementation.

Table 2: **Test accuracy (%) of a WRN-16-4 model trained from scratch:** Our proposed DP-MIX$_{\text{SELF}}$ technique significantly outperforms De et al. [9] Self-Aug method (baseline and prior SoTA) in all privacy budget settings and all three datasets considered.

| Dataset | Method | $\varepsilon = 1$ | $\varepsilon = 2$ | $\varepsilon = 4$ | $\varepsilon = 8$ |
|---|---|---|---|---|---|
| CIFAR-10 | Self-Aug | 56.8 (.5) | 62.9 (.3) | 69.5 (.4) | 78.7 (.5) |
| | DP-MIX$_{\text{SELF}}$ | **57.2** (.4) | **64.6** (.4) | **70.47** (.4) | **79.8** (.3) |
| CIFAR-100 | Self-Aug | 13.3 (.5) | 20.9 (.4) | 31.8 (.2) | 39.2 (.4) |
| | DP-MIX$_{\text{SELF}}$ | **14.1** (.4) | **21.5** (.4) | **33.3** (.3) | **40.6** (.3) |
| EuroSAT | Self-Aug | 75.4 (.3) | 81.1 (.1) | 85.8 (.2) | 89.7 (.3) |
| | DP-MIX$_{\text{SELF}}$ | **75.7** (.2) | **82.8** (.3) | **87.4** (.2) | **90.8** (.2) |

### 5.2 Finetuning with DP

We consider two pretrained models Vit-B-16 and ConvNext. As baselines/prior SoTA we consider De et al. [9] and the Mixed Ghost Clipping technique of Bu et al. [5]. Results are shown in Table 3.

We observe that our proposed DP-MIX$_{\text{SELF}}$ method yields significant improvements over all prior methods across datasets and privacy budgets. We also observe that DP-MIX$_{\text{DIFF}}$, which uses samples from the diffusion model, significantly outperforms DP-MIX$_{\text{SELF}}$ on datasets such as Caltech256, SUN397, and Oxford-IIIT Pet. Notably, when the privacy budget is limited, such as $\varepsilon = 1$, we observe remarkable improvements compared to DP-MIX$_{\text{SELF}}$ (e.g., $8.5\%$ on Caltech256 using Vit-B-16). This shows the benefits of incorporating diverse images from the diffusion model via mixup.

On the other hand, we observe that diffusion examples do not provide a benefit for datasets such as CIFAR-10 and EuroSAT. We investigate the cause of this empirically in Appendix B (supplementary materials). On these datasets, DP-MIX$_{\text{DIFF}}$ only sometimes outperforms the prior baselines, but DP-MIX$_{\text{SELF}}$ provides a consistent improvement, suggesting that samples from the diffusion model

Table 3: **Test accuracy (%) of fine-tuned Vit-B-16 and ConvNext**: We fine-tune Vit-B-16 and ConvNext models on CIFAR-10, CIFAR-100, EuroSAT, Caltech256, SUN397 and Oxford-IIT Pet datasets using different $\varepsilon$ with $\delta = 10^{-5}$, and report the test accuracy and standard deviation. Our proposed methods, DP-MIX$_{\text{SELF}}$ and DP-MIX$_{\text{DIFF}}$, outperform the baselines in all cases.

| Dataset | Method | Vit-B-16 | | | | ConvNext | | | |
|---|---|---|---|---|---|---|---|---|---|
| | | $\varepsilon = 1$ | $\varepsilon = 2$ | $\varepsilon = 4$ | $\varepsilon = 8$ | $\varepsilon = 1$ | $\varepsilon = 2$ | $\varepsilon = 4$ | $\varepsilon = 8$ |
| CIFAR-10 | Mixed Ghost | 95.0 (.1) | 95.0 (.2) | 95.3 (.4) | 95.3 (.2) | 94.6 (.1) | 94.6 (.1) | 94.7 (.0) | 94.7 (.1) |
| | Self-Aug | 96.5 (.1) | 97.0 (.0) | 97.1 (.0) | 97.2 (.0) | 95.9 (.0) | 96.4 (.1) | 96.5 (.1) | 96.5 (.0) |
| | DP-MIX$_{\text{SELF}}$ | **97.2** (.3) | **97.4** (.2) | **97.4** (.2) | **97.6** (.3) | **96.8** (.1) | **96.9** (.1) | **96.9** (.1) | **97.3** (.1) |
| | DP-MIX$_{\text{DIFF}}$ | 97.0 (.2) | 97.1 (.1) | 97.2 (.1) | 97.3 (.1) | 96.3 (.1) | 96.5 (.1) | 96.6 (.1) | 96.7 (.1) |
| CIFAR-100 | Mixed Ghost | 78.2 (.4) | 78.5 (.1) | 78.4 (.3) | 78.4 (.1) | 74.9 (.3) | 75.1 (.1) | 75.5 (.2) | 75.8 (.1) |
| | Self-Aug | 79.3 (.2) | 83.2 (.3) | 83.5 (.3) | 84.2 (.1) | 75.8 (.3) | 80.0 (.1) | 81.4 (.2) | 82.2 (.2) |
| | DP-MIX$_{\text{SELF}}$ | 81.8 (.2) | 83.5 (.1) | **84.5** (.1) | **84.6** (.2) | 78.2 (.3) | 80.9 (.2) | **82.3** (.1) | **82.3** (.1) |
| | DP-MIX$_{\text{DIFF}}$ | **82.0** (.1) | **83.8** (.1) | 84.0 (.1) | 84.3 (.1) | **79.4** (.2) | **81.4** (.2) | 81.6 (.1) | 81.8 (.2) |
| EuroSAT | Mixed Ghost | 84.0 (.1) | 84.8 (.2) | 84.9 (.1) | 85.0 (.2) | 85.5 (.2) | 86.9 (.1) | 87.0 (.6) | 87.6 (.3) |
| | Self-Aug | 93.3 (.2) | 94.1 (.2) | 95.4 (.2) | 95.5 (.2) | 93.5 (.2) | 94.5 (.4) | 95.2 (.1) | 95.2 (.1) |
| | DP-MIX$_{\text{SELF}}$ | **94.3** (.1) | **94.9** (.2) | **95.6** (.2) | **95.6** (.1) | **94.6** (.1) | **94.7** (.1) | **95.4** (.2) | **95.5** (.1) |
| | DP-MIX$_{\text{DIFF}}$ | 92.6 (.1) | 93.4 (.1) | 93.9 (.2) | 93.9 (.1) | 92.8 (.2) | 93.2 (.1) | 93.6 (.2) | 93.8 (.1) |
| Caltech256 | Mixed Ghost | 79.7 (.2) | 88.2 (.2) | 91.4 (.2) | 92.3 (.2) | 79.2 (.2) | 87.4 (.2) | 88.0 (.1) | 88.2 (.2) |
| | Self-Aug | 80.4 (.1) | 89.7 (.2) | 92.0 (.1) | 93.2 (.2) | 80.0 (.2) | 88.2 (.1) | 91.0 (.1) | 92.2 (.1) |
| | DP-MIX$_{\text{SELF}}$ | 81.2 (.2) | 90.1 (.2) | 92.2 (.2) | 93.4 (.1) | 81.0 (.1) | 88.7 (.2) | 91.3 (.1) | 92.4 (.1) |
| | DP-MIX$_{\text{DIFF}}$ | **89.7** (.2) | **91.8** (.2) | **92.9** (.1) | **93.9** (.1) | **88.7** (.2) | **91.8** (.2) | **92.6** (.2) | **93.3** (.2) |
| SUN397 | Mixed Ghost | 70.7 (.2) | 71.2 (.1) | 72.2 (.2) | 72.5 (.2) | 64.3 (.2) | 65.0 (.2) | 65.3 (.1) | 65.3 (.1) |
| | Self-Aug | 72.7 (.1) | 76.0 (.1) | 78.0 (.1) | 79.6 (.2) | 72.2 (.1) | 76.5 (.1) | 78.0 (.2) | 78.9 (.1) |
| | DP-MIX$_{\text{SELF}}$ | 73.2 (.1) | 76.5 (.2) | 78.7 (.2) | 79.6 (.1) | 72.5 (.1) | 76.8 (.1) | 78.5 (.0) | 79.5 (.1) |
| | DP-MIX$_{\text{DIFF}}$ | **75.1** (.2) | **77.8** (.1) | **79.5** (.2) | **80.6** (.1) | **75.0** (.2) | **77.5** (.1) | **79.3** (.1) | **80.0** (.1) |
| Oxford Pet | Mixed Ghost | 71.2 (.2) | 79.1 (.2) | 80.4 (.2) | 81.0 (.2) | 65.2 (.2) | 78.2 (.2) | 79.1 (.2) | 79.9 (.2) |
| | Self-Aug | 72.2 (.2) | 82.1 (.2) | 85.8 (.3) | 88.2 (.1) | 68.1 (.2) | 81.3 (.2) | 85.5 (.1) | 87.0 (.1) |
| | DP-MIX$_{\text{SELF}}$ | 72.5 (.2) | 82.5 (.2) | 86.8 (.2) | 88.7 (.2) | 68.8 (.2) | 81.7 (.2) | 86.3 (.2) | 87.7 (.2) |
| | DP-MIX$_{\text{DIFF}}$ | **83.2** (.3) | **86.3** (.2) | **88.3** (.2) | **89.4** (.2) | **80.5** (.2) | **86.2** (.2) | **88.2** (.2) | **88.8** (.2) |

can inhibit performance. On CIFAR-100, DP-MIX$_{\text{DIFF}}$ outperforms other methods only at low privacy budgets ($\varepsilon \leq 2$), with DP-MIX$_{\text{SELF}}$ providing the best performance in the high privacy budget regime.

# 6 Ablation Study: Why Does DP-MIX Improve Performance?

We perform ablation experiments to better understand why our methods consistently and significantly outperform the prior SoTA techniques.

## 6.1 Understanding Self-Augmentations

Since DP-MIX$_{\text{SELF}}$ does not alter the label and involves applying mixup to (augmentations of) a single original training example, the method could be viewed as just another single-sample data augmentation technique. The question then becomes what augmentation techniques improve performance when using differential privacy and why.

De et al. [9] use flipping and cropping as self-augmentation (Self-Aug). We perform experiments using several widely used augmentation techniques (based on Chen et al. [7] — Colorjitter, Translations and Rotations, Cutout, Gaussian Noise, Gaussian Blur, and Sobel Filter) in addition to Self-Aug to compare them to DP-MIX$_{\text{SELF}}$. We set $K_{\text{BASE}} = 16$ as in previous experiments. Results are shown in Table 4, where the second to last column combines all augmentations together (randomly chosen). We observe that these augmentations do **not** enhance performance to the same degree as DP-MIX$_{\text{SELF}}$. In a few cases, the augmentations improve performance slightly above Self-Aug, but some of them also diminish performance.

## 6.2 Number of Self-Augmentations

Increasing the number of self-augmentations beyond 16 does not improve performance [9]. This was also pointed out recently by Xiao et al. [41]. In fact, sometimes increasing the number of

Table 4: **Test accuracy (%) of Self-Aug with other Augmentations:** Vit-B-16 model performance on CIFAR-10 and CIFAR-100 with $\varepsilon = 1$ with different augmentations. We observe that these augmentations do not enhance performance to the same degree as DP-MIX$_{\text{SELF}}$.

| Dataset | Self-Aug | +Jitter | +Affine | +Cutout | +Noise | +Blur | +Sobel | +All | DP-MIX$_{\text{SELF}}$ |
|---|---|---|---|---|---|---|---|---|---|
| CIFAR-10 | 96.5 (.1) | 96.3 (.1) | 96.2 (.1) | 96.5 (.1) | 95.8 (.2) | 96.7 (.2) | 72.5 (.1) | 96.5 (.2) | **97.2 (.3)** |
| CIFAR-100 | 79.3 (.2) | 78.1 (.2) | 73.9 (.2) | 79.6 (.1) | 73.5 (.2) | 80.0 (.1) | 9.5 (.2) | 79.4 (.1) | **81.8 (.2)** |

self-augmentations hurts performance. We perform experiments by varying the number of self-augmentations ($K_{\text{BASE}}$) on CIFAR-10. Results are shown in Table 5. We observe that $K_{\text{BASE}} = 16$ is optimal for the self-augmentations proposed by De et al. [9]. However, we obtain significantly better performance DP-MIX$_{\text{SELF}}$ for $K = 32$ ($K_{\text{BASE}} = 16$, $K_{\text{SELF}} = 16$ and $K_{\text{DIFF}} = 0$). Recall that setting $K_{\text{DIFF}} = K_{\text{SELF}} = 0$ in Algorithm 1 recovers the Self-Aug method of [9].

Table 5: **Increase $K_{\text{BASE}}$ from 16 to 36 for Self-Aug.** We conduct experiments on CIFAR-10 with $\varepsilon = 8$ and $\delta = 10^{-5}$ in two settings: train a WRN-16-4 model from scratch and fine-tune a pre-trained Vit-B-16. We can observe that for both cases, there are no substantial performance improvements when increasing $K_{\text{BASE}}$.

| Training method | Total # of Aug | WRN-16-4 | Vit-B-16 (pretrained) |
|---|---|---|---|
| Self-Aug | 16 | 78.8 (.5) | 97.2 (.0) |
| | 24 | 78.5 (.4) | 97.0 (.1) |
| | 32 | 78.4 (.3) | 97.0 (.1) |
| | 36 | 78.6 (.4) | 97.0 (.1) |
| DP-MIX$_{\text{SELF}}$ | 32 | **79.8** (.3) | **97.6** (.3) |

## 6.3 Pretraining or DP-MIX$_{\text{DIFF}}$?

Since DP-MIX$_{\text{DIFF}}$ uses synthetic samples from a diffusion model, we consider pretraining the model on the synthetic samples $\mathcal{D}$ from the diffusion model with SGD prior to fine-tuning with DP. The results, presented in Table 6, indicate that pretraining on $\mathcal{D}$ does not improve the model's performance. This indicates that the improved performance of DP-MIX$_{\text{DIFF}}$ results from the way it uses the diffusion samples, not just from having access to diffusion samples.

Table 6: **Test accuracy (%) with and without pretraining on synthetic diffusion samples**: We compare the performance of fine-tuned Vit-B-16 and ConvNext with DP on Caltech256 with versus without pretraining on $\mathcal{D}$ (using SGD). We set $\varepsilon = 2$ and $\delta = 10^{-5}$.

| Pretrained on $\mathcal{D}$ | Vit-B-16 | ConvNext |
|---|---|---|
| Yes | 81.5 (.0) | 80.9 (.1) |
| No (Ours) | **91.8 (.2)** | **91.8 (.2)** |

## 6.4 Effect of DP-MIX on Training Data

In Table 7, we compare the distributions of the effective train sets produced by different data augmentation techniques against those of the original train and test sets, i.e., we measure the Fréchet Inception Distance (FID), where lower FID values indicate more similarity between the datasets. Note, the FIDs are consistent with test accuracy across datasets and methods. For example, the FID of Self-Aug + Colorjitter is significantly larger than FID of Self-Aug, explaining why adding Colorjitter decreases test accuracy: it results in data less similar to train and test sets. Similarly, the FID for DP-MIX$_{\text{DIFF}}$ for EuroSAT is much larger compared to the other datasets, which explains reduced test accuracy relative to the other methods. In Table 8, we compare the FID between the original train and test sets with the FID between the original train set and the synthetic samples generated by the text-to-image diffusion model that we employ for DP-MIX$_{\text{DIFF}}$. Note the much greater FID between the train set and the diffusion samples. This large domain gap may explain why, as shown in Table 6,

Table 7: **FID for different methods' generated images** compared to training set and test set.

| Dataset | Train | | | | Test | | | |
|---|---|---|---|---|---|---|---|---|
| | Self-Aug | +Jitter | DP-MIX$_{\text{SELF}}$ | DP-MIX$_{\text{DIFF}}$ | Self-Aug | +Jitter | DP-MIX$_{\text{SELF}}$ | DP-MIX$_{\text{DIFF}}$ |
| CIFAR-10 | 2.6 | 6.0 | 3.1 | 3.3 | 3.1 | 6.4 | 3.5 | 3.7 |
| CIFAR-100 | 3.1 | 6.1 | 3.3 | 3.5 | 3.5 | 6.5 | 3.8 | 3.9 |
| EuroSAT | 2.2 | 6.1 | 3.0 | **6.6** | 4.1 | 8.3 | 6.1 | **10.9** |
| Caltech256 | 1.0 | 2.9 | 1.3 | 1.5 | 2.5 | 4.2 | 2.8 | 2.9 |

Table 8: **FID between datasets for different FID models**. We compute FID values between the training set and test set, and training set and text-to-images diffusion model generated images based on different FID models (InceptionV3 and Vit-B-16).

| Dataset | InceptionV3 | | Vit-B-16 | |
|---|---|---|---|---|
| | Train vs Test | Train vs Diffusion | Train vs Test | Train vs Diffusion |
| CIFAR-10 | 3.2 | 30.5 | 0.4 | 30.1 |
| CIFAR-100 | 3.6 | 19.8 | 0.5 | 21.9 |
| EuroSAT | 7.4 | **164.0** | 7.3 | **82.9** |
| Caltech256 | 6.8 | 25.1 | 1.6 | 37.4 |

pretraining on the diffusion samples result in much worse performance compared to our proposed DP-MIX$_{\text{DIFF}}$.

## 6.5 Influence of $K_{\text{BASE}}$, $K_{\text{SELF}}$ and $K_{\text{DIFF}}$

Recall, $K_{\text{BASE}}$ is the number of base self-augmentations, $K_{\text{SELF}}$ is the number of mixups, and $K_{\text{DIFF}}$ is the number of synthetic diffusion samples used. We vary $K_{\text{BASE}}$, $K_{\text{SELF}}$ and $K_{\text{DIFF}}$ values and report their performance in Table 9. We observe that selecting $K_{\text{DIFF}} = 2$ or $K_{\text{DIFF}} = 4$ and setting $K_{\text{SELF}} \leq K_{\text{BASE}}$ gives good results overall. In this paper, we used $K_{\text{BASE}} = 16$, $K_{\text{SELF}} = 18$ and $K_{\text{DIFF}} = 2$ for most cases as it provides good overall results across many datasets and settings [3].

Table 9: **Influence of $K_{\text{BASE}}$, $K_{\text{SELF}}$ , and $K_{\text{DIFF}}$ for fine-tuning Vit-B-16 model on Caltech256** with $\varepsilon = 1$ and $\delta = 10^{-5}$.

| $K_{\text{BASE}}$ | 8 | 8 | **8** | 8 | 8 | **8** | 8 | 8 | 8 | 16 | 16 | 16 | 16 | 16 |
|---|---|---|---|---|---|---|---|---|---|---|---|---|---|---|
| $K_{\text{SELF}}$ | 8 | 8 | **8** | 16 | 16 | **16** | 24 | 24 | 24 | 8 | 8 | 8 | 16 | 16 |
| $K_{\text{DIFF}}$ | 0 | 2 | **4** | 0 | 2 | **4** | 0 | 2 | 4 | 0 | 2 | 4 | 0 | 2 |
| **Acc. (%)** | 80.4 | 85.0 | **87.2** | 77.1 | 84.5 | **87.2** | 77.2 | 84.2 | 86.4 | 77.7 | 84.2 | 86.7 | 81.2 | 83.8 |

| $K_{\text{BASE}}$ | 16 | 16 | 16 | 16 | 24 | 24 | 24 | 24 | 24 | 24 | 24 | 24 | 24 |
|---|---|---|---|---|---|---|---|---|---|---|---|---|---|
| $K_{\text{SELF}}$ | 16 | 24 | 24 | 24 | 8 | 8 | 8 | 16 | 16 | 16 | 24 | 24 | 24 |
| $K_{\text{DIFF}}$ | 4 | 0 | 2 | 4 | 0 | 2 | 4 | 0 | 2 | 4 | 0 | 2 | 4 |
| **Acc. (%)** | 86.3 | 77.3 | 82.6 | 85.8 | 77.9 | 82.7 | 85.5 | 77.9 | 82.1 | 85.1 | 80.1 | 81.5 | 84.5 |

## 7 Conclusions

We studied the application of multi-sample data augmentation techniques such as mixup for differentially private learning. We show empirically that the most obvious way to apply mixup, using microbatches, does not yield models with low generalization errors as microbatching is at an inherent disadvantage. We then demonstrate how to harness mixup without microbatching by applying it to self-augmentations of a single training example. This provides a significant performance increase over the prior SoTA. Finally, we demonstrate that producing some augmentations using text-to-image diffusion models further enhances performance when combined with mixup.

---

[3]We use larger $K_{\text{DIFF}}$ i.e. 24 for Caltech256 and Oxford Pet as it provides better performance.

## Acknowledgments

This work was supported in part by the National Science Foundation under CNS-2055123. Any opinions, findings, and conclusions or recommendations expressed in this material are those of the authors and do not necessarily reflect the views of the National Science Foundation.

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

# A Algorithm: DP-SGD

---

**Algorithm 2** DP-SGD from Abadi et al. [2]

---

**Require:** Training data $x_1, ..., x_N$, loss function $\mathcal{L}(\theta) = \frac{1}{N} \sum_i \mathcal{L}(\theta, x_i)$. Parameters: learning rate $\eta_t$, noise scale $\sigma$, group size $B$, gradient norm bound $C$.

   **Initialize** $\theta_0$ randomly

   **for** $t \in [T]$ **do**

      Take a random sample $B_t$ with sampling probability $B/N$

      **for** each $i \in B_t$ **do**

         Compute $\mathbf{g}_t(x_i) \leftarrow \nabla_\theta \mathcal{L}(\theta_t, x_i)$

         $\bar{\mathbf{g}}_t(x_i) \leftarrow \mathbf{g}_t(x_i)/\max(1, \frac{\|\mathbf{g_t(x_i)}\|_2}{C})$

      **end for**

      $\tilde{\mathbf{g}}_t \leftarrow \frac{1}{B} \sum_i (\bar{\mathbf{g}}_t(x_i) + \mathcal{N}(0, \sigma^2 C^2 \mathbf{I}))$

      $\theta_{t+1} \leftarrow \theta_t - \eta_t \tilde{\mathbf{g}}_t$

   **end for**

**Ensure:** $\theta_T$ and compute the overall privacy cost $(\varepsilon, \delta)$ using a privacy accounting method

---

# B Additional Experimental Results

## B.1 How useful is the synthetic data?

Let $\mathcal{S}_1$ and $\mathcal{S}_2$ denote equally sized disjoint subsets of the private dataset, and $\mathcal{D}$ an equally sized set of synthetic samples generated via diffusion, as described in Section 3.1.

For this experiment, we consider three different training methods.

- **Method A**: Fine-tune on $\mathcal{S}_1$ using DP-MIX$_{\text{SELF}}$.
- **Method B**: Fine-tune on $\mathcal{S}_1$ using DP-MIX$_{\text{DIFF}}$ with mixup images sampled from $\mathcal{D}$.
- **Method C**: Same as Method B, but replace set $\mathcal{D}$ with set $\mathcal{S}_2$. Note, we assume that $\mathcal{S}_2$ is publicly available, so accessing it during training does not incur a privacy cost.

The intuition behind this experiment is that method C provides an upper bound for both A and B, since, in the best case, the distribution of the synthetic data would exactly match that of the private data. The experimental results, presented in Table 10, validate this intuition.

For CIFAR-100, Caltech256, SUN397, and Oxford Pet, method C outperforms method A. Consistent with this, method B, our proposed DP-MIX$_{\text{DIFF}}$, also provides a performance boost. On the other hand, for CIFAR-10 and EuroSAT, method C, despite directly using private data for mixup, does not meaningfully improve performance. Similarly, method B also does not improve performance. For EuroSAT, method B slightly decreases performance, due to the large domain gap (large FID) between the synthetic and private data.

Table 10: **Test accuracy (%) using different training methods** with Vit-B-16 and ConvNext on various datasets after fine-tuning. We set the $\varepsilon = 2$ and $\delta = 10^{-5}$.

| Dataset | Vit-B-16 | | | ConvNext | | |
|---|---|---|---|---|---|---|
| | **A** | **B** | **C** | **A** | **B** | **C** |
| CIFAR-10 | 96.9 (.1) | 96.9 (.0) | 97.0 (.1) | 96.2 (.1) | 96.1 (.1) | 96.2 (.1) |
| CIFAR-100 | 81.3 (.3) | 82.0 (.2) | 82.1 (.1) | 76.7 (.3) | 78.0 (.1) | 78.4 (.1) |
| EuroSAT | 93.7 (.1) | 91.4 (.1) | 93.9 (.2) | 93.8 (.2) | 92.3 (.1) | 93.9 (.2) |
| Caltech 256 | 76.0 (.4) | 81.9 (.1) | 82.3 (.2) | 76.4 (.3) | 81.1 (.1) | 81.6 (.3) |
| SUN397 | 70.8 (.2) | 72.4 (.1) | 73.8 (.1) | 70.1 (.2) | 72.3 (.1) | 73.6 (.2) |
| Oxford Pet | 65.1 (.3) | 68.3 (.1) | 68.3 (.1) | 64.6 (.2) | 66.0 (.1) | 68.1 (.1) |

## B.2 Reproducing De et al. [9] and DP-MIX$_{\text{SELF}}$ in JAX

For completeness, we train WRN-16-4 on CIFAR10 using Self-Aug and our proposed DP-MIX$_{\text{SELF}}$ using [9]'s official JAX code (`https://github.com/deepmind/jax_privacy`). As in [9], we repeat each experiment 5 times and report median test accuracy in Table 11. The results are consistent

with those presented in the rest of our paper — DP-MIX$_{\text{SELF}}$ outperforms Self-Aug for different privacy budgets.

Table 11: **Performance comparison on JAX.**

| Method | $\varepsilon = 1$ | $\varepsilon = 8$ |
|---|---|---|
| Self-Aug | 56.3 (.3) | 79.4 (.1) |
| DP-MIX$_{\text{SELF}}$ | 57.1 (.4) | 80.0 (.2) |

## B.3 Pure-DP-MIX$_{\text{DIFF}}$

To demonstrate the influence of $K_{\text{BASE}}$ in our method, we set $K_{\text{BASE}} = 0$ and call it Pure-DP-MIX$_{\text{DIFF}}$. In effect Pure-DP-MIX$_{\text{DIFF}}$ is simply mixing up the synthetic examples themselves. We test it on CIFAR-100 and represent it in Table 12. We can see that Pure-DP-MIX$_{\text{DIFF}}$ offers much worse performance than both DP-MIX$_{\text{SELF}}$ and DP-MIX$_{\text{DIFF}}$, although it still offers better performance than Self-Aug due to the beneficial effects of mixup. More generally, we think that Pure-DP-MIX$_{\text{DIFF}}$ will tend to worsen an overfitting problem whenever there is a large domain gap between the original training data and the diffusion samples. DP-MIX$_{\text{DIFF}}$ does not suffer from this problem because it ensures that (augmented versions) of the original training data samples are seen during training.

Table 12: **Test accuracy (%) of Pure-DP-MIX$_{\text{DIFF}}$ ($K_{\text{BASE}} = 0$) on CIFAR-100 with Vit-B-16 model.** We set $\delta = 10^{-5}$ and $\varepsilon = 1$. We can observe that Pure-DP-MIX$_{\text{DIFF}}$ does not improve performance, which shows the necessity of using base augmentations ($K_{\text{BASE}} > 0$).

| Method | Test accuracy |
|---|---|
| Self-Aug | 79.3 (.2) |
| DP-MIX$_{\text{SELF}}$ | 81.8 (.2) |
| DP-MIX$_{\text{DIFF}}$ | **82.0** (.1) |
| Pure-DP-MIX$_{\text{DIFF}}$ | 80.9 (.2) |

## B.4 Running time

We provide the running time for different methods in Table 13. All experimental runs utilized a single A100 GPU and were based on the same task of finetuning the Vit-B-16 model on the Caltech256 dataset for 10 epochs. Due to additional augmentation steps, the training time of our methods is longer than prior work.

Table 13: **Running time** for different methods of the same task (fine-tuning Vit-B-16 on Caltech256 for 10 epochs). We use one A100 GPU for each training method.

| Method | Self-Aug | DP-MIX$_{\text{SELF}}$ | DP-MIX$_{\text{DIFF}}$ |
|---|---|---|---|
| Running time | 2h 12min | 7h 33min | 7h 40 min |

## B.5 Effect of Mixup on Gradients

We study what happens to gradients and parameter updates during training for our methods versus Self-Aug. Fig. 2 plots the per-parameter gradient magnitude averaged over samples at each epoch (prior to clipping and noise adding). The histogram shows the data averaged over all training epochs and the X% color lines show that data only for the epoch at X% of the total training process. There are 10 epochs for this experiment – for example, the line for $20\%$ shows the data for epoch 2.

The figure shows more concentrated values for our methods compared to the Self-Aug baseline, which suggests more stable training and faster convergence. Standard deviations for CIFAR-10 with Self-Aug, DP-MIX$_{\text{SELF}}$ and DP-MIX$_{\text{DIFF}}$ are: $2.16 \cdot 10^{-3}$, $9.76 \cdot 10^{-4}$ and $9.59 \cdot 10^{-4}$, respectively. For Caltech256 they are: $1.43 \cdot 10^{-3}$, $1.07 \cdot 10^{-3}$ and $9.32 \cdot 10^{-4}$, respectively. This is consistent with experimental results of test accuracies for each method.

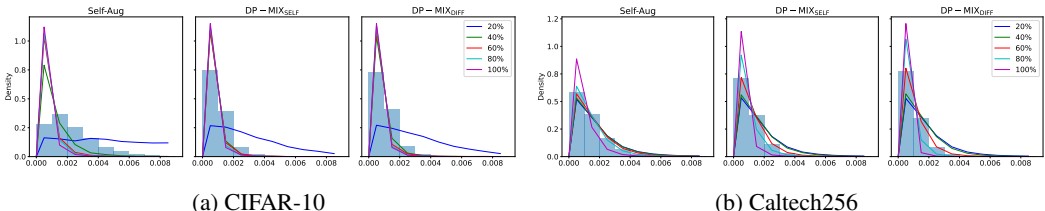

(a) CIFAR-10                               (b) Caltech256

Figure 2: **Per-parameter gradient magnitude before clipping and adding noise** on CIFAR-10(a) and Caltech256(b) with fine-tuning Vit-B-16 model with $\varepsilon = 1$ and $\delta = 10^{-5}$. The different curves represent different training stages. Values for our proposed DP-$\text{MIX}_{\text{SELF}}$ and DP-$\text{MIX}_{\text{DIFF}}$ are more concentrated suggesting more stable training and faster convergence.

## C    Additional Experimental Details

**Datasets**    We use the following datasets:

- **CIFAR-10** is a widely-utilized dataset in the field of computer vision, serving as a standard for evaluating image recognition algorithms. Collected by Alex Krizhevsky, Vinod Nair, and Geoffrey Hinton [21], this dataset is a crucial tool for machine learning research. The dataset consists of 60,000 color images, each sized at $32 \times 32$ pixels, and categorized into 10 different classes like cats, dogs, airplanes, etc. We use 50,000 data points for training, and 10,000 for the test set.
- **CIFAR-100** is a well-regarded dataset in the domain of computer vision, typically used for benchmarking image classification algorithms. This dataset, also collected by Krizhevsky et al. [21], is a key resource in machine learning studies. CIFAR-100 comprises 60,000 color images, each of 32x32 pixel resolution. What distinguishes it from CIFAR-10 is the higher level of categorization complexity; the images are sorted into 100 distinct classes instead of 10. We use 50,000 data points for training, and 10,000 for the test set.
- **EuroSAT** [18] is a benchmark dataset for deep learning and image classification tasks. This dataset is composed of Sentinel-2 satellite images spanning 13 spectral bands and divided into ten distinct classes. It has 27,000 labeled color images which size is $64 \times 64$. We use 21600 as the training set and 5400 as a test set.
- **Caltech 256** [15]. Caltech is commonly used for image classification tasks and comprises of 30607 RGB images of 257 different object categories. For our experiments, we designated 80% of these images for training and the remaining 20% for testing.
- **SUN397** The Scene UNderstanding (SUN) [42, 43] database contains 108,754 RGB images from 397 classes. For our experiments, we use 80% of these images for training and the remaining 20% for testing.
- **Oxford Pet** [28] contains 37 classes of cats and dogs and we use 3680 images for training and the rest 3669 images for testing.

For all our experiments, we maintained the clip norm at $C = 1$, with the exception of Mix-ghost clipping where we used $C = 0.05$ as required by the original paper [5] and its implementation[4]. The noise level was automatically calculated by Opacus based on the batch size, target $\varepsilon$, and $\delta$, as well as the number of training epochs.

**Implementation details of Section 3.**    In this experiment, we set $K_{\text{BASE}} = 16$ and adjust the hyperparameters according to the recommendations in the original paper [9]. To facilitate the implementation in PyTorch and enable microbatch processing, we adapt our code based on two existing code bases [5].

**Implementation details of Section 5.1.**    In training our models from scratch, we adhere to a $K_{\text{BASE}}$ value of 16 and adjust the other hyperparameters in accordance with the guidelines provided in the original papers [9, 5]. For our fine-tuning experiments, we maintain the same $K_{\text{BASE}}$ value and perform hyperparameter searches in each case to ensure we utilize the optimal learning rate. Importantly, we

---

[4]`https://github.com/JialinMao/private_CNN`

[5]`https://github.com/facebookresearch/tan` and `https://github.com/ChrisWaites/pyvacy`

do not incorporate any learning rate schedules, as per the suggestions of other research papers [5, 9] that indicate such schedules do not yield performance improvements.

**Implementation details of Section 5.2 and Appendix B.** In order to generate synthetic samples, we feed the text prompt `'a photo of a <class name>'` to the diffusion model [13]. For each dataset, the number of synthetic samples we generate is equal to the number of real images in the dataset.

**Source Code.** Readers can find further experimental and implementation details in our open-source code at `https://github.com/wenxuan-Bao/DP-Mix`.

# D   Ethical Considerations & Broader Impacts

In this paper, we propose an approach to use multi-sample data augmentation techniques such as mixup to improve the privacy-utility tradeoff of differentially-private image classification. Since differential privacy offers strong guarantees, its deployment when training machine learning has the potential to substantially reduce harm to individuals' privacy. However, some researchers have observed that although the guarantee of differential privacy is a worst-case guarantee, the privacy obtained is not necessarily uniformly spread across all individuals and groups. Further, there is research suggesting that differential privacy may (in some cases) increase bias and unfairness, although these findings are disputed by other research.

