# OpenReview forum: "DP-Mix: Mixup-based Data Augmentation for Differentially Private Learning"
_NeurIPS.cc/2023/Conference — NeurIPS 2023 poster_

### Official Review · Reviewer_DhpR · 2023-06-27

**Soundness:** 2 fair
**Presentation:** 3 good
**Contribution:** 2 fair
**Rating:** 5
**Confidence:** 3

**Summary:**

The paper proposes two data augmentation techniques, MIXUP-SELFAUG and MIXUP-DIFFUSION, for differentially private learning. The authors investigate why naive applications of multi-sample data augmentation techniques, such as mixup, fail to achieve good performance and propose these two techniques specifically designed for the constraints of differentially private learning. MIXUP-SELFAUG performs mixup on self-augmented data, which results in lower gradients norm average and variance leading to smoother training. MIXUP-DIFFUSION further improves performance by incorporating synthetic data from a pre-trained diffusion model into the mixup process. The paper also discusses the challenges of applying data augmentation techniques in the context of differential privacy and compares the proposed techniques with existing methods on various datasets. The experimental results show that the proposed techniques achieve state-of-the-art classification performance across a range of datasets and settings. Overall, the paper proposed effective data augmentation techniques that can improve the performance of machine learning models while preserving the privacy of sensitive data.



**Strengths:**

This paper is easy to read. This paper considerately used DP-SGD for differentially private learning. These techniques improve the performance of mixup-based data augmentation. The paper also discusses the challenges of applying data augmentation techniques in the context of differential privacy.

**Weaknesses:**

1. The absence of a theoretical analysis for MIXUP-SELFAUG in the submitted work is notable. The authors have provided an intuition on why mixup could harm differential privacy and presented an alternative approach to mitigate this issue. It could further strengthen the paper if the authors could provide a theoretical guarantee for differential privacy.


2. In relation to MIXUP-SELFAUG which doesn't mix labels, questions arise about why MIXUP-SELFAUG improves performance. The original intent of mixup, which is label interpolation for better performance, seems not fully addressed in this context. The authors could consider explaining their approach's effectiveness via intuition or mathematical formulation.

[1] Zhang, Linjun, et al. "How does mixup help with robustness and generalization?." ICLR 2021

[2] Jeong, Jongheon, et al. "Smoothmix: Training confidence-calibrated smoothed classifiers for certified robustness." NeurIPS 2021

[3] Zhang, Linjun, et al. "When and how mixup improves calibration." ICML 2022

[4] Park, Chanwoo, Sangdoo Yun, and Sanghyuk Chun. "A unified analysis of mixed sample data augmentation: A loss function perspective." NeurIPS 2022

[5] Zou, Difan, et al. "The benefits of mixup for feature learning." arXiv preprint arXiv:2303.08433 (2023).

[6] Oh, Junsoo, and Chulee Yun. "Provable Benefit of Mixup for Finding Optimal Decision Boundaries." ICML 2023


3. Concerning the implementation of the diffusion model, there are a few considerations that the authors might need to address. The necessity for incorporating a diffusion model is not sufficiently justified in the current draft. For instance, the diffusion model used in this study is pretrained, which could potentially introduce the element of knowledge distillation. This raises the question: Is the observed performance improvement mainly due to distillation effects? Additionally, the use of a diffusion model not trained with DP-SGD could potentially pose issues for differential privacy learning. The authors are encouraged to provide clarification on these points.


4. While the adoption of a diffusion model might benefit differential privacy, the paper could delve deeper into explaining why it's advantageous for ensuring differential privacy. A detailed discussion on this matter could provide readers with a better understanding of the authors' choice and its implications on the overall study. The authors' insights on this matter would be greatly appreciated.

[1] Chourasia, Rishav, Jiayuan Ye, and Reza Shokri. "Differential privacy dynamics of langevin diffusion and noisy gradient descent." NeurIPS 2021

[2] Dockhorn, Tim, et al. "Differentially private diffusion models." arXiv preprint arXiv:2210.09929 (2022).

[3] Ghalebikesabi, Sahra, et al. "Differentially Private Diffusion Models Generate Useful Synthetic Images." arXiv preprint arXiv:2302.13861 (2023).

[4] Lyu, Saiyue, et al. "Differentially Private Latent Diffusion Models." arXiv preprint arXiv:2305.15759 (2023).

**Questions:**

I want to clarify especially the second and third issues of the weakness. If this is well-addressed, I am down to re-evaluate this paper.

**Limitations:**

The authors adequately addressed the limitation and potential negative societal impact on their work.

---

> ### Author Rebuttal · Authors · 2023-08-10
>
> Thank you for the constructive feedback and detailed comments. We will correct typographical errors and make suggested improvements to our paper. Here are our clarifications:
>
> **1. The absence of a theoretical analysis for MIXUP-SELFAUG in the submitted work is notable.**\
> Our work is largely empirical in nature. However, theoretical justification for our approach can be provided with respect to why naive application of mixup harms DP and our approach does not.
>
> Consider Alg. 1 in the submission PDF. The inner loop is what satisfies DP and it does so for a batch of example $B_t$. (The overall privacy guarantee is obtained using amplification theorems and composition as explained in related work [2,3] from the submission PDF.)
>
> Let $B = \{ (x_1, y_1), (x_2, y_2), \ldots, (x_m, y_m) \}$ denote a batch of $m=|B|$ examples. Consider the clipped-gradients sum, defined as: $$ \bar{g}(B) = \sum_i {g}(z_i)/\max(1, \frac{||{g}(z_i)||_2}{C}) $$
>     where $z_i = (x_i, y_i)$ is the $i^{\rm th}$ example of the batch and $g(\cdot)$ denotes the gradient of the loss on an example with respect to the model parameters. The $L_2$ sensitivity of a function $f$ is denoted by $\Delta f$ and is defined as:
>     $$
>     \Delta f = \max || f(B) - f(B') ||_2,
>     $$
>     where we take the maximum over pairs of neighboring batches $B, B'$.
>
> DP-SGD relies on the clipping step ${g}(z_i)/\max(1, \frac{||{g}(z_i)||_2}{C})$ to bounds the sensitivity of the sum to $C$. If we add or remove any example $(x, y)$ to/from $B$ the $L_2$-norm of only one term changes and because it is clipped to norm $C$, the clipped-gradients sum can change by at most $C$.
>
> What if we now apply mixup to the batch $B$? To apply mixup to the batch $B$ we may randomly pair up examples in $B$ (e.g., ignoring any leftover example) and for each pair we create a single mixup example according to Eq. (4) in the submission PDF. In that case the new mixup-augmented batch contains $m$ original examples plus $\lfloor \frac{m}{2} \rfloor$ mixed-up examples, one for each pair of original examples. We then apply DP-SGD as in Alg. 1 to this augmented batch. We denote this construction applied to a batch $B$ as $\rm{mixup}(B)$.
>
>  We provide a proof sketch (due to space constraints) that the sensitivity of the clipped-gradients sum of $\rm{mixup}(B)$ is $2C$ (and not $C$).
>
> (Proof sketch.) Suppose we apply $\rm{mixup}$ on the batch $B$ to obtain a mixup-augmented batch and then compute the clipped-gradients sum on that augmented batch. an original example $z \in B$ will impact two terms of the sum (the one involving the gradient of $z$ and also the one involving the gradient of the mixed-up pair involving $z$). For example this can happen if every term of the sum not involving $z$ points to some direction $C \vec{e}$ for some $\vec{e}$ and the two terms involving $z$ point to $-C\vec{e}$. Therefore the sensitivity is $2C$.
>
> The sensitivity being $2C$ (instead of $C$) means that the scale of Gaussian noise added must be doubled to obtain the same privacy guarantee.
>
> Our methods do not suffer from this issue (sensitivity remains $C$) because clipping occurs on the gradients of the ``microbatch'' formed by all of the augmentations (only one term of the clipped-gradients sum of norm at most $C$ is added/removed).
> \
> \
> **2. In relation to MIXUP-SELFAUG which doesn't mix labels, questions arise about why MIXUP-SELFAUG improves performance.**\
> Our intuition is that data augmentation and mixup help generalization. It's also worth pointing out that mixup does provide a benefit even without knowledge/use of true labels as evidence by works that apply mixup outside of the supervised settings. For example, see: [link1](https://arxiv.org/pdf/1905.04215.pdf), [link2](https://arxiv.org/pdf/2206.07692.pdf) and [link3](https://arxiv.org/pdf/2108.12296.pdf). The last paper is particularly relevant because they apply mixup on pairs of examples with the *same* label.
> \
> \
> **3. The necessity for incorporating a diffusion model is not sufficiently justified in the current draft.**\
> We discuss this in the general response. We made sure to select a diffusion model in a way that would not inadvertently bias results.
> \
> \
> **4. Is the observed performance improvement mainly due to distillation effects?**\
> We don't believe so. First, we aligned the training data for the diffusion models and pre-trained models to ensure that Mixup-Diffusion would not have access to data (through the synthetic examples) that other methods did not also have. Second, we explicitly tested whether the synthetic examples from the diffusion model by themselves explain the observed performance. For this, we fine-tuned the model with DP-SGD with and without pretrained on the synthetic examples. Results in supplementary materials (Table 6 --- A.2) show that pre-training with the synthetic examples provides does not improve performance. This shows that the benefits come from mixup and not distillation effects.
> \
> \
> **5. The use of a diffusion model not trained with DP-SGD could potentially pose issues for differential privacy learning.**\
> As we explain in the general response and above, the diffusion model and pre-trained models used the same training set. This means that there is no data leaked through the diffusion model that was not also used to pre-train the fine-tuned model.
> \
> \
> **6. Why it's advantageous for ensuring differential privacy.**\
> We believe DP-SGD benefits from seeing more data during training and that's what mixup with synthetic diffusion provides.

---

> ### Comment · Reviewer_DhpR · 2023-08-10
> **Thank you.**
>
> Thank you for your detailed explanation.
>
> Regarding the first point, I appreciate your clarification. However, I am not sure that the sensitivity difference between 2C and C might not be significant given that it's a constant factor. Plus, it is not a worst-case guarantee, so it would be more clarified.
>
> For the second point, I might need further clarity. In the third link you shared, they utilized a "contrastive" loss to find a good representation. However, it appears this paper does not employ the same approach. I'm curious to understand how this work might benefit from the same label without using contrastive loss.
>
> As for points 3, 4, and 5, my concerns remain. My understanding is that the diffusion model isn't a "differential private" method since it hasn't been trained with DP-SGD. Could you explain this? If a model isn't trained using a "differential private" method, wouldn't it naturally exhibit improved performance? I agree with your point on the alignment of training data and the testing with a synthetic dataset.
>
> I'm not sure that the absence of data leaks isn't necessarily the same as ensuring differential privacy. (Using the same dataset also might induce not good differential privacy. Not really related I believe)
>
> Thank you once again for your time and dedication in addressing my queries.

---

> > ### Author Response · Authors · 2023-08-11
> >
> > Thank you for your quick response. We would like to clarify some points:
> >
> >
> >
> > **1.**  The difference between C and 2C is massive because it implies effectively doubling the noise level to achieve the same privacy guarantee with DP-SGD, which results in much worse performance. For example, see Table 2 of our submission PDF for EuroSAT, where for $\varepsilon=1$ and $\varepsilon=2$ correspond to almost doubling the noise level (to $\sigma=25$ from $\sigma=13$) with all other parameters (batch size, number of epochs, etc.) remaining the same (except the learning rate which is tuned to give best results in each case). As can be seen in Table 2, the absolute test accuracy decrease resulting from this almost doubling of the noise level is over $5$%.
> >
> > Also, to clarify: our analysis for the doubling of sensitivity is a worst-case analysis. This is because differential privacy is a worst-case notion, as it applies to all pairs of neighboring datasets.
> >
> >
> >
> > **2.**  Here we want to clarify two separate points:
> >
> > - **a.** Most data augmentation techniques, such as flipping and cropping, operate on a single example and thus do not mix labels, yet they still improve generalization. Mixup-SelfAug operates in a similar fashion.
> >
> > - **b.** The related works mentioned in our previous response are not meant to endorse any particular approach or suggest that we would adopt it. Our point is simply that empirical results (from these related works) show that mixup without label mixing still enhances generalization.
> >
> >
> > **3.** Thank you for clarifying. We think we understand what you are asking now.
> >
> > If we train a model with DP-SGD, the differential privacy protection is always with respect to the training set used. This means that if we take a model already pre-trained on dataset X and then fine-tune it with DP-SGD on dataset Y, the differential privacy guarantee only applies to dataset Y; there is no guarantee on dataset X unless pre-training was also done using DP-SGD.
> >
> > This is why in our paper we consider both training from scratch and fine-tuning. In the training from scratch setting, we consider only Mixup-SelfAug and not Mixup-Diffusion. In the fine-tuning/pre-trained setting, we assume that the dataset used to pre-train the model (LAION-2B) is public so DP-SGD pre-training is not necessary. This is a standard assumption from prior works. For example, De et al. [10] pretrain their models on ImageNet, JFT-300M, and JFT-4B. In fact, the paper that introduced DP-SGD, Abadi et al. [2], pretrain their model on CIFAR-100.
> >
> > In our case, we specifically pretrain ViT and ConvNext on the LAION-2B dataset so that both the model to be fine-tuned and the diffusion model used the exact same training data. Given the assumption that this dataset is public, we do not need to do the pretraining with DP-SGD. When we fine-tune the pretrained model with Mixup-SelfAug or Mixup-Diffusion differential privacy protects the fine-tuning data.

---

> > > ### Comment · Reviewer_DhpR · 2023-08-11
> > > **Thank you**
> > >
> > > For the first point, I understand that there might be a significant difference between C and 2C. Thank you for clarifying that.
> > >
> > > Regarding the second point, I don't necessarily agree. The purpose of mixup is to blend both labels and data. If we don't blend the labels, there would be no reason to use mixup; instead, we'd just employ traditional augmentation techniques like flipping. Mixup is done for smoothing the trained function so label mixing is necessary to have these effects. And smoothing trained function actually provides good generalization.  (See Figure 1 in https://arxiv.org/pdf/1710.09412.pdf) My contention is that the effects observed might not stem from the "mixup" method per se, but could be attributed to having a large dataset or incorporating translations and rotations. How can we clarify this distinction?
> > >
> > > I've fully grasped the third point in the sense that previous works also use pre-trained models, while do not fully agree that this is a valid approach. But, For point 3, I think this is enough to discuss (it is a really fundamental question to the literature). Thank you for the detailed explanation.

---

> > > > ### Author Response · Authors · 2023-08-16
> > > >
> > > > **Is mixup without label mixing similar to traditional augmentation?**
> > > >
> > > > We agree with the reviewer that it is important to consider whether mixup without mixing labels could have a similar effect as other traditional augmentations such as translations and rotations. Therefore, we performed additional experiments using several widely-used augmentation techniques ([https://arxiv.org/abs/2002.05709](https://arxiv.org/abs/2002.05709) -- Colorjitter, Translations and Rotations, Cutout, Gaussian Noise, Gaussian Blur, and Sobel Filter) in addition to Self-Aug to compare them to Mixup-SelfAug. The table below shows the results for each augmentation separately with the last row combining all augmentations together (randomly chosen). We observe that these augmentations do *not* enhance performance to the same degree as Mixup-SelfAug. In a few cases, the augmentations improve performance slightly above Self-Aug, but some of them also diminish performance. We are thankful for the suggestion and will add this table to the revised paper.
> > > >
> > > > **Table:** Vit-B-16 model performance on CIFAR-10 and CIFAR-100 with $\varepsilon$=1.
> > > >
> > > > | Method                                | CIFAR-10 | CIFAR-100 |
> > > > |---------------------------------------|----------|-----------|
> > > > | Self-Aug                              | 96.49% (±0.12%)   | 79.28%(±0.18%)    |
> > > > | Mixup-SelfAug                         | 97.21%(±0.27%)   | 81.75% (±0.15%)    |
> > > > | Self-Aug + Colorjitter               | 96.30% (±0.09%)  | 78.15% (±0.21%)    |
> > > > | Self-Aug + Translations and rotations | 96.17%(±0.14%)   | 73.87% (±0.18%)    |
> > > > | Self-Aug + Cutout                     | 96.50% (±0.10%)  | 79.55% (±0.11%)    |
> > > > | Self-Aug + Gaussian Noise             | 95.84%(±0.21%)   | 73.47%(±0.17%)    |
> > > > | Self-Aug + Gaussian Blur              | 96.72%(±0.16%)   | 79.95% (±0.13%)   |
> > > > | Self-Aug + Sober Filter               | 72.47%(±0.11%)   | 9.54% (±0.22%)    |
> > > > | Self-Aug + All (Random)               | 96.45%(±0.15%)   | 79.42% (±0.14%)    |

---

> > > > > ### Comment · Reviewer_DhpR · 2023-08-16
> > > > > **Quick question**
> > > > >
> > > > > So these results are all about using the same number of samples?
> > > > > Thank you very much. I will adjust my score.

---

> > > > > > ### Author Response · Authors · 2023-08-17
> > > > > >
> > > > > > Yes, for all Self-Aug we use k=16 and for Mixup-SelfAug k=k'=16. Regarding the number difference between Self-Aug and Mixup-SelfAug, please check point 3 of our rebuttal and Table 6 of our attached one page pdf,  it shows that increasing k cannot improve performance for Self-Aug.

---

> > > > > > > ### Comment · Reviewer_DhpR · 2023-08-17
> > > > > > > **Thank you**
> > > > > > >
> > > > > > > I changed my score again because now every point is clear. Even though this cannot make the theoretical explanation (about why mixing the same label helps), I think this holds empirically by various experiments that the authors provided.

---

### Official Review · Reviewer_sSKD · 2023-07-04

**Soundness:** 3 good
**Presentation:** 4 excellent
**Contribution:** 3 good
**Rating:** 5
**Confidence:** 5

**Summary:**

This paper considers the privacy-utility tradeoff for ML models trained with differential privacy guarantees, and develops a technique using data augmentation on image datasets to train models with high accuracies on standard benchmarks with DP guarantees. Mixup, a commonly used augmentation technique in computer vision, and its variants are not compatible with standard DP-SGD because a single datapoint could influence many training instances through augmentations. Recent work [10] pointed out that augmentations involving a single datapoint can be used if clipping if done after averaging all gradients stemming from augmentations of the datapoint. The present paper develops two methods inspired by mixup that involve a single datapoint and therefore can be used with DP-SGD. Experiments show performance comparable to state-of-the-art methods for training DP models.

**Strengths:**

Thank you to the authors for sharing their interesting research! Overall, the paper tackles an important subject - improving the utility of models with rigorous privacy guarantees. This is an active area of research, and the paper engages with recent work appropriately. The proposed methods are simple to implement for image datasets and provide a boost in utility. An attempt is made to explain why mixup methods improve performance by checking the distributions of gradient norms.

Originality: The paper develops two variants of mixup that can be applied to DP-SGD training. I do not know of prior work using mixup for DP-SGD.

Quality: The analysis of differential privacy is careful and correct. The authors are diligent in highlighting parts of the analysis which are particularly tricky (L91, etc.) although these aspects are known in the literature. Relevant and recent methods are used for benchmarking, and error bars over 3 runs are given. Source code is provided.

Clarity: The paper is quite clear.

Significance: The poor utility of models trained with DP-SGD holds back the adoption of this important privacy enhancing technology, so new methods (like the ones in this paper) that improve the utility are necessary.

**Weaknesses:**

Originality: The paper combines the augmentation techniques developed for DP in [10], with the idea of mixup which is very standard. The combination is only slightly non-trivial in this context.

Quality: The experiments lack detail - it would not be possible to reproduce the author's work with the current state of the paper and appendices. Prior SotA work [10] is frequently referenced, but performance levels from that paper are not reproduced or surpassed. The experiment using diffusion models is not benchmarked fairly. Broader impacts are relegated to the appendix.

Clarity: There are minor grammatical and typesetting issues, but they do not hurt the clarity of the paper. Figure 2 requires clarification.

Significance: The applicability of this technique is limited to image datasets, whereas that of [10] for example can be applied much more broadly.

**Questions:**

1. What augmentations were used in the experiments? I have not found this detail clearly stated in the paper or appendices, despite it being a central focus. L167 mentions a "randomized transformation function $T$" but this is not brought up again. L247 mentions flipping and cropping, but it seems to imply these are not the only augmentations used. The lack of important details such as this negatively impact the reproducibility of this research.

2. Can the authors state what computing resources were used and information on the runtime of their methods in comparison to prior work?

3. Can the authors give details on the diffusion model that was used (architecture, how it was trained, etc.)? All we know is that it was from Open Clip and trained on LAION-2B (L218).

4. The experiments attempt to follow [10] in many ways for close comparison, but the results reported in Section 5 as baselines do not reach the level of that paper. For instance, [10] reports 81.4% accuracy on CIFAR-10 training from scratch with a WRN-16-4 and epsilon=8. Table 1's Baseline is presumably meant to represent the method from [10], but only achieves 72.5% accuracy. The author's proposed method achieves 78.7% accuracy in the same setting. Given that the authors make claims of SotA performance, can they clarify why the results reported in [10] were not directly used or reproduced?

5. There are some results which are repeated across tables (e.g. first two rows of Table 4 are reproduced from Table 3). Could this be mentioned explicitly where applicable so that it is easier to track and compare results? A reader moving quickly might assume they are separate results.

6. In Table 4 the Mixup-Diffusion method uses additional data compared to the other approaches, namely the data generated by the diffusion model. Since the utility of DP models is often constrained by access to data, it is important to use baselines that have the same level of data access. Have the authors put thought into an appropriate baseline technique in the style of DP-SGD that also makes use of the synthetic data? (The most naive option being to pre-train or extend the training dataset with the same synthetic data made available to the mixup-diffusion method.)

7. In Figure 2 it is not clear what the difference is between the curves and the blue bars. I interpret each curve as the distribution of gradient norms over individual datapoints in the training dataset at a particular epoch (these details are not given, I have to guess). However, the blue bars are stated to be a histogram of the average norm of gradients, but what is the average taken over and what is the resulting distribution over? I note that the paper at various times considers epochs, minibatches, and microbatches, so the authors should clearly state which notion they are using in their analysis.

8. Also for Figure 2, the authors do not mention in the paper or appendices which dataset is used. Can the authors provide sufficient detail to reproduce their experiments (without having to dig through source code)?

9. The result in Figure 2 showing lower average gradient norms is indeed suggestive that mixup leads to smoother training and faster convergence. However, have the authors found an explanation for why mixup leads to lower average gradient norms? This is not at all self-evident, especially for Mixup-Diffusion where training and synthetic datapoints are mixed, and those images can come from different classes (as illustrated in Fig 1).

10. Have the authors checked the equivalent of Figure 2 for Mixup-Diffusion, or for other datasets? Can this analysis be included?

Minor:
In L67, couldn't $\delta=0$ be a valid choice, and for that matter $\delta=1$ corresponding to no privacy guarantee?

In L58, typically one chooses $\delta$ to be much less than $|D|^{-1}$, but the authors state $\delta=o(|D|)$.

Incomplete list of small errors:

L70 add -> adds

L72 gradient -> gradients

L89 vector the -> vector of the

Alg 2 line 2, missing space

L150 extra parens ((x_1, y_1)

**Limitations:**

The proposed method is implicitly stated to only apply to image datasets, but this limitation is not openly addressed by the authors. Similar work used for benchmarking is not limited in the same way.

The paper focuses on the utility of private models, but does not mention that private training can have negative effects other than reducing utility (other than in a brief Appendix B without references to prior work). For instance, it is well-known that DP-SGD usually exacerbates the biases in models making them more unfair [A][B][C]. Does data augmentation affect fairness? What are the biases introduced from the synthetic data? Robustness is another topic aligned with Trustworthy ML, and researchers often introduce augmented data during training to improve it. It is possible that the proposed method has positive effects on robustness, but the authors have not addressed this direction despite the significant literature on the intersection of privacy and robustness.

[A] Bagdasaryan et al. 'Differential privacy has disparate impact on model accuracy' NeurIPS 2019

[B] Xu et al. 'Removing disparate impact on model accuracy in differentially private stochastic gradient descent' ACM SIGKDD 2021

[C] Esipova et al. 'Disparate impact in differential privacy from gradient misalignment' ICLR 2023

----
Summary of Discussion: I have read all reviews and rebuttals for this submission.

The authors responded to all points of my review. My rating was maintained at 5 as some of my original points remain as concerns, including the originality, and quality (lack of details of reproduction).

---

> ### Author Rebuttal · Authors · 2023-08-10
>
> Thank you for the constructive feedback and detailed comments. We will correct typographical errors and make suggested improvements to our paper. Here are our clarifications:
> \
> \
> **1. The paper combines the augmentation techniques developed for DP in [10].**\
> We agree with the reviewer that we exploit the insights of [10] to maintain the privacy guarantee. However, the novelty of our paper is that we show how to exploit the augmentation technique of [10] for mixup. The benefit of the self-augmentations proposed by De et al. [10] quickly hits diminishing returns ($k=16$ -- see Table 6 in the attached 1-page rebuttal PDF) and adding different types of augmentations also does not help. By contrast, Mixup-Diffusion and Mixup-SelfAug provide consistently better performance by leveraging the benefits of mixup.
> \
> \
> **2. The applicability is limited to image datasets, whereas that of [10] for example can be applied much more broadly.**\
> We acknowledge that we only investigated our techniques for image data (similar to De et al. [10]). However, we disagree that our techniques are necessarily limited to image data. There are data augmentation techniques for other data domains and numerous related work showing successful application of mixup to other data domains (e.g., see (https://arxiv.org/pdf/2108.12296.pdf), (https://arxiv.org/pdf/1905.08941.pdf), (https://arxiv.org/pdf/2010.02394.pdf)). Some of these techniques apply mixup in the feature/embedding space instead of the input data. But, this is in principle compatible with our techniques and gradient clipping over all augmentations would also preserve the privacy guarantee in such cases. We leave for future work the investigation of whether the same benefits are obtained.
> \
> \
> **3. What augmentations were used?**\
> We apologize for any confusion. To clarify, our methods employ the exact same augmentations as De et al. [10] (flipping and cropping).
> \
> \
> **4. Can the authors state what computing resources and information on the runtime of their methods in comparison to prior work?**\
> We provide the running time for different methods in the table below. All experimental runs utilized a single A100 GPU and were based on the same task of finetuning the Vit-B-16 model on the Caltech256 dataset for 10 epochs. Due to additional augmentation steps the training time of our methods is larger than prior work.
>
>  **Table:** Running time for different methods of the same task (fine-tuning Vit-B-16 on Caltech256 for 10 epochs). We use one A100 GPU for each training method.
>
>  | Method       | Self-Aug | Mixup-SelfAug | Mixup-Diffusion |
>  |--------------|----------|---------------|-----------------|
>  | Running time | 2h 12min | 7h 33min      | 7h 40min        |
>
> \
> **5. Details on the diffusion model that was used? All we know is that it was from Open Clip and trained on LAION-2B (L218).**\
> We provide more information in the general response. But to be clear the diffusion model is stable-diffusion-v1-4 (https://huggingface.co/CompVis/stable-diffusion-v1-4). The models from Open Clip are Vit-B-16 and ConvNext (which we pre-trained on LAION-2B the same data used to train the diffusion model).
> \
> \
> **6. Baselines do not reach the level of that paper**\
> We think there is a misunderstanding. The WRN-16-4 model with $\varepsilon=8$, [10] indicates a performance of 79.5% on CIFAR-10 (as can be confirmed in Table 3 of their paper). This closely aligns with our reported performance of 78.74%.
>
> Additionally, it's crucial to highlight the specifications in our submission PDF Table 1: ``Baseline'' in that table refers to vanilla DP-SGD without Self-Aug or any modifications. Elsewhere we sometimes refer to Self-Aug as baseline, since it is the prior SoTA.
>
> To clarify: both of our methods Mixup-SelfAug and Mixup-Diffusion consistently outperform the Self-Aug [10] baseline in both pre-trained and from scratch settings. For example, Mixup-SelfAug obtains $79.83$% test accuracy in the setting where Self-Aug obtains $78.74$% (and [10] reports $79.5$%). The point of Table 1 in our paper is to show why microbatching will not work for mixup.
> \
> \
> **7. There are some results that are repeated across tables.**\
> Yes, we will revise this.
> \
> \
> **8. Have the authors put thought into an appropriate baseline technique in the style of DP-SGD that also makes use of the synthetic data?**\
> Yes, we conducted such experiments in the supplementary material (A.2 --- Table 6). We pretrain the model on the synthetic data, which is an appropriate baseline because if the performance boost we get could be obtained using the synthetic data directly (i.e., without mixup) then it may not make sense to use Mixup-Diffusion. However, pretraining on the synthetic data does *NOT* help. This implies that the benefit of Mixup-Diffusion is due to how it combines synthetic data into the mixup augmentations.
> \
> \
> **9. Regarding Figure 2**\
> We have reproduced the experiments and clarified this in the general response.
> \
> \
> **10. Have the authors found an explanation for why mixup leads to lower average gradient norms?**\
> We do not claim that mixup leads to lower average gradient norms. Rather we observe from Figure 2 (and Figure 1 in the attached 1-page PDF) that gradients for mixup are more concentrated (i.e., lower variance/std). See the general response for more details.
> \
> \
> **11. In L67, couldn't $\delta=0$ be a valid choice, and for that matter $\delta=1$ corresponding to no privacy guarantee?**\
> Yes we will fix this. $\delta=0$ or $\delta=1$ are allowable.
> \
> \
> **12. Does data augmentation affect fairness? What are the biases introduced from the synthetic data? Robustness?**\
> This is a question worth investigating. We believe it is outside the scope of our current paper, so we leave it for future work.

---

> > ### Comment · Reviewer_sSKD · 2023-08-14
> > **Response to rebuttals**
> >
> > I have read all the reviews and rebuttals.
> >
> > 6. I was looking at the WRN-40-4 rather than WRN-16-4 in [10], so this is clear now.
> >
> > I was unclear about what "Baseline" referred to in Table 1, this could be made more explicit in the work.
> >
> > However, the SelfAug's reported performance of 79.5% [10] is significantly different from your number, 78.7% in Table 2, given that Mixup-SelfAug achieves 79.8%. At this level, the approaches are within one standard deviation on performance.
> >
> > 10. Does Fig 2 not show that the average norm is reduced? If the distribution is more concentrated around zero, wouldn't the average norm be reduced?

---

> > > ### Author Response · Authors · 2023-08-17
> > >
> > > Thank you for your insights and feedback.
> > >
> > > 1. We will clarify the meaning of “Baseline” in Table1 of the revised paper.
> > >
> > > 2. To ensure a fair comparison, we employ Sander et al.’s (ICML 2023, https://openreview.net/pdf?id=DIkGgI9baJ) reproduction of [10] for both SelfAug and Mixup-SelfAug.
> > >
> > > 3.  Fig 2. does indeed show that the average norm is reduced – there's a notable reduction from approximately 0.009 to 0.0004 for CIFAR-10 and 0.001 to 0.0003 for Caltech256.

---

> > > > ### Author Response · Authors · 2023-08-20
> > > >
> > > > For completeness, we train WRN-16-4 on CIFAR10 using Self-Aug and our proposed Mixup-SelfAug using [10]’s official JAX code (https://github.com/deepmind/jax_privacy). As in [10], we repeat each experiment 5 times and report median test accuracy in the table below. The results are consistent with those presented in our paper – Mixup-SelfAug outperforms Self-Aug for different privacy budgets.
> > > >
> > > > | Method       | $\varepsilon = 1$            | $\varepsilon=8$            |
> > > > |--------------|------------------|------------------|
> > > > | Self-Aug     | 56.28%(±0.30%)   | 79.42%(±0.14%)   |
> > > > | Mixup-SelfAug| 57.05%(±0.35%)   | 80.02%(±0.18%)   |

---

### Official Review · Reviewer_GjeG · 2023-07-06

**Soundness:** 3 good
**Presentation:** 4 excellent
**Contribution:** 3 good
**Rating:** 6
**Confidence:** 4

**Summary:**

The paper studies mixup data augmentation for differentially private (DP) machine learning. The traditional mixup data augmentation requires multiple samples. Therefore, applying them in DP model training is not straightforward. The paper first shows that a naive way of implementing it with micro-batches of size 2 gives poor results. To improve the performance, the paper proposes two new approaches. The first approach mixes up a sample and its own augmented version. The second approach mixes up a sample with generated samples from diffusion models. Experiments show that the proposed approaches improve over self-augmentation.

**Strengths:**

* The paper is well-written.

* Mixup data augmentation is very useful in the non-DP world. However, incorporating that into DP training is non-trivial. The paper is a valuable step towards bringing the benefit of mixup data augmentation to DP training.

**Weaknesses:**

* I have some doubts about whether the experimental comparison is fair.

* More ablation studies are needed.

See "questions" for the details.


**Questions:**


My major concern is the experimental comparison between self-aug and the proposed mixup variants in Tables 2 and 3.

* In the experiments of Table 2, did we keep the number of augmented samples the same? To make it more accurate, let's consider Mixup-SelfAug and use the notations in Section 3.2. For a fair comparison, we need to make sure that the "k" utilized in self-aug equals "k+k'" used in Mixup-SelfAug. (The motivation is to keep the computation cost roughly the same.)

* Similarly, for Table 3, did we keep the number of augmented samples the same?

Other questions:

* It is better to show how k, k', and k'' influence the performance of Mixup-SelfAug and Mixup-Diffusion.

* Is there any benefit of combining Mixup-SelfAug and Mixup-Diffusion together?

I will update the score according to the answers to the above questions.

Other minor issues:

* Figure 2: I do not fully understand the figure. It says "The histogram shows average norm of gradients.", but I do not get which values are averaged over. Is the process (1) for each sample, computing the average norms of gradients across all iterations, and (2) plotting the histogram of the above values of all samples? Similarly, how are the lines computed?
* Line 138: "however" -> ". However"
* Line 150: "((x_1,y_1)" -> "(x_1, y_1)"
* Line 249-254: The information in this paragraph is also covered in the method section.


**Limitations:**


The paper did not discuss the limitation or the potential negative societal impact.

---

> ### Author Rebuttal · Authors · 2023-08-10
>
> Thank you for the constructive feedback and detailed comments. We will correct typographical errors and make suggested improvements to our paper. Here are our clarifications:
> \
> \
> **1. The experimental comparison between self-aug and the proposed mixup variants in Tables 2 and 3.**\
> For Self-Aug, increasing the number of augmentations $ k $ beyond $ k=16 $ does not increase (and often decreases) performance. We show this in Section 5.3 of our paper and the original paper from De et al. [10] shows the same phenomenon. To further address your concerns, we have conducted an additional experiment where we increase $ k $ from $ 16 $ to $ 36 $ for both 'training from scratch' and 'pretrain' settings. Results are shown in Table 6 (attached 1-page PDF). The best performance for Self-Aug is $ k=16 $. Furthermore, for the same number of augmentations Mixup-SelfAug outperforms Self-Aug. For $ k+k'=32 $ augmentations, Mixup-SelfAug also outperforms Self-Aug with $ k=36 $ augmentations. This shows that our comparisons are indeed fair. In fact, we give an advantage to Self-Aug in our paper because we report results for that method with the value of $ k $ (i.e, $ k=16 $) that gives the best results for it.
> \
> \
> **2. How $ k $, $ k' $, and $ k'' $ influence the performance of Mixup-SelfAug and Mixup-Diffusion.**\
> We have conducted additional experiments to show this. See the table below. Recall: $ k $ is the number of base self-augmentations, $ k' $ is the number of mixups, and $ k'' $ is the number of synthetic diffusion samples used. Overall, selecting $ k''=2 $ or $ k''=4 $ and setting $ k' $&le;$ k $ gives good overall results. In our paper, we used $ k=k'=16 $ and $ k''=2 $ as it provides good overall results across many datasets and settings.
> \
> \
> **Table:** Change $ k $, $ k' $, and $ k'' $ for fine-tuning Vit-B-16 model on Caltech 256 with  $\varepsilon = 1$  and  $\delta = 10^{-5}$. Here $ k $ is the number of augmented data, $ k' $ is the number of mixup processes, and $ k'' $ is the number of diffusion samples.
>
> | $ k $ | $ k' $ | $ k'' $ | Acc. (%) | Rank |
> |---|---|---|---|---|
> | 8  | 8  | 0 | 80.39 | 20 |
> | 8  | 8  | 2 | 85.03 | 9  |
> | 8  | 8  | 4 | 87.21 | 1  |
> | 8  | 16 | 0 | 77.05 | 27 |
> | 8  | 16 | 2 | 84.45 | 11 |
> | 8  | 16 | 4 | 87.19 | 2  |
> | 8  | 24 | 0 | 77.16 | 26 |
> | 8  | 24 | 2 | 84.23 | 12 |
> | 8  | 24 | 4 | 86.35 | 4  |
> | 16 | 8  | 0 | 77.66 | 24 |
> | 16 | 8  | 2 | 84.22 | 13 |
> | 16 | 8  | 4 | 86.71 | 3  |
> | 16 | 16 | 0 | 81.21 | 19 |
> | 16 | 16 | 2 | 83.76 | 14 |
> | 16 | 16 | 4 | 86.28 | 5  |
> | 16 | 24 | 0 | 77.28 | 25 |
> | 16 | 24 | 2 | 82.55 | 16 |
> | 16 | 24 | 4 | 85.80 | 6  |
> | 24 | 8  | 0 | 77.94 | 22 |
> | 24 | 8  | 2 | 82.72 | 15 |
> | 24 | 8  | 4 | 85.54 | 7  |
> | 24 | 16 | 0 | 77.89 | 23 |
> | 24 | 16 | 2 | 82.10 | 17 |
> | 24 | 16 | 4 | 85.08| 8  |
> | 24 | 24 | 0 | 80.14 | 21 |
> | 24 | 24 | 2 | 81.46 | 18 |
> | 24 | 24 | 4 | 84.54 | 10 |
>
> \
> **3. Is there any benefit of combining Mixup-SelfAug and Mixup-Diffusion together?**\
> First, note that Mixup-SelfAug can be seen as a special case of Mixup-Diffusion where $ k''=0 $. We introduced Mixup-Diffusion to improve upon Mixup-SelfAug by introducing additional data diversity through diffusion samples. To investigate your specific question, we conduct an experiment using what we call ``Pure-Mixup-Diffusion'', i.e., setting $k=0$ for Mixup-Diffusion, meaning that the original training samples are not used at all. In effect Pure-Mixup-Diffusion is simply mixing up the synthetic examples themselves. The table below compares Pure-Mixup-Diffusion to other methods. We can see that Pure-Mixup-Diffusion offers much worse performance than both Mixup-SelfAug and Mixup-Diffusion, although it still offers better performance than Self-Aug due to the beneficial effects of mixup. More generally, we think that Pure-Mixup-Diffusion will tend to worsen an overfitting problem whenever there is a large domain gap between the original training data and the diffusion samples. Mixup-Diffusion does not suffer from this problem because it ensures that (augmented versions) of the original training data samples are seen during training.
>
> **Table:** Implement Pure-Mixup-Diffusion (K=0) on CIFAR-100 with Vit-B-16 model. We set  $\delta = 10^{-5}$  and $\varepsilon = 1$ . We can observe that Pure-Mixup-Diffusion cannot improve the model's performance.
>
> | Method               | Test accuracy                |
> |----------------------|------------------------------|
> | Self-Aug             | 79.28% (±0.18%)              |
> | Mixup-SelfAug        | 81.75% (±0.15%)              |
> | Mixup-Diffusion      | **82.02%** (±0.11%)          |
> | Pure-Mixup-Diffusion | 80.91% (±0.17%)              |
>
> \
> **4. Explanation of Figure 2 is not so clear.**\
> We acknowledge that our explanations lacked clarity and details. We have produced new figures and provided clarification of this in the general response.
> \
> \
> **5. The paper did not discuss the limitation or the potential negative societal impact.**\
> We briefly discuss potential negative societal impact in the supplementary material (see Appendix B).

---

> > ### Comment · Reviewer_GjeG · 2023-08-16
> >
> > The rebuttal fully addressed my concerns. Therefore, I increase the score.

---

### Official Review · Reviewer_ayCk · 2023-07-06

**Soundness:** 3 good
**Presentation:** 2 fair
**Contribution:** 2 fair
**Rating:** 4
**Confidence:** 4

**Summary:**

This paper proposes a data augmentation technique for differentially private deep learning using mixup regularization. Mixup is a popular augmentation technique which involves taking linear combinations of training samples to create new samples. However, such a technique cannot be directly applied to differentially private training, as the sensitivity of the private algorithm increases rendering it to be un-useful. Towards this end the authors propose two methods for private mixup augmentation. First, is mixup with self augmentation which involves mixing samples obtained from augmentation of a single sample, and secondly, mixing training samples with samples generated from diffusion models. The paper provides empirical results to support their claim.

**Strengths:**

Mixup is an interesting technique, and exploring it for differentially private training is important.
The technique is simple to implement in practice with easy to prove theoretical guarantees.
Method shows improved empirical performance on the presented datasets.


**Weaknesses:**

The proposed method isnt too different from the method proposed by De et al which is cited in the paper. Its a minor augmentation to the augmentation they proposed.
Pre-trained models are trained on large datasets but fine-tuning is performed only on toy datasets. Needs more empirical evidence on practical datasets (for example, oxfordpets, flowers, and so on)


**Questions:**

For diffusion model based experiments the pre-trained model is trained on 2B images which is likely to contain the entire fine-tuning sets. Does this seem like a reliable public pre-training to choose?
Can you increase the number of samples to mixup from 2? How does the performance change by increasing the number of samples in the convex combination?
What should be the nature of the distribution of samples that are used during mixup? Can out of distribution samples be mixed to obtain meaningful results?


**Limitations:**

The method is only restricted to self augmentations or augmentations with publicly available data. In private training having different gradients aligned when training with DP-SGD is important to reduce the effect of the gaussian noise added to ensure DP. Will it be possible to construct a mixup involving multiple samples (rather than self-mixup or diffusion model based) so that the gradients during training are more aligned, which thereby increase the signal-to-noise ratio?

---

> ### Author Rebuttal · Authors · 2023-08-10
>
> Thank you for the constructive feedback and detailed comments.
> \
> \
> **1. The proposed method isn't too different from the method proposed by De et al.**\
> Our work uses the same insight as De et al. [10] --- by clipping the gradients aggregated over all the augmentations we preserve the DP guarantee. However, the same idea naively applied to mixup (or multi-sample data augmentation) does not work. Our contribution is showing how to make it work and that it yields substantial benefits.
> \
> \
> **2. Pre-trained models are trained on large datasets but fine-tuning is performed only on toy datasets.**\
> We extended our evaluation to include three additional datasets: Caltech256, SUN397, and Oxford-IIIT Pet, some of which we already used in supplementary material. See Table 1 in the attached rebuttal PDF. Our proposed methods significantly outperform the current SoTA ("Self-Aug" --- De et al. [10]) across these new datasets and for all values of $\varepsilon$. It is also worth pointing out that prior SoTA (De et al. [10]) not only uses some of the same datasets (e.g., CIFAR-10 and CIFAR-100) but also pre-trains models on very large datasets including ImageNet, JFT-4B, and JFT-300M.
> \
> \
> **3. Pre-trained model is trained on 2B images which is likely to contain the entire fine-tuning sets.**\
> No, we don't believe so. As we explain in the general response, we took great care in selecting the pre-trained models and pre-training data to ensure fair comparisons. There are two important points: First, even if there was overlap between the pre-training data and the fine-tuning sets our methods still outperform Self-Aug when fine-tuning the *exact same pre-trained model*. From this we can conclude that our method provides a boost beyond whatever advantage there could be from the pre-trained models' training data. Second, we show in supplementary material A.2 (Table 6) that fine-tuning the models using synthetic data from a diffusion model trained on the exact same 2B images dataset does *not* improve performance. If the fine-tuning data were included in the diffusion model training data, wouldn't we see substantial performance boost from training on synthetic examples from that very diffusion model?
> \
> \
> **4. Can you increase the number of samples to mixup from 2? How does the performance change by increasing the number of samples in the convex combination?**\
> Yes, we increase the number of samples to mixup from 2 to 3 and 4, and present its performance on CIFAR-100 with $\varepsilon=1$, in the table below. Increasing the number of samples for Mixup-Diffusion does not improve accuracy.
>
>
> | # of samples for Mixup | Test accuracy       |
> |------------------------|---------------------|
> | 2                      | 82.02% (±0.11%)     |
> | 3                      | 81.35% (±0.15%)     |
> | 4                      | 81.33% (±0.09%)     |
>
> \
> **5. What should be the nature of the distribution of samples that are used during mixup? Can out of distribution samples be mixed to obtain meaningful results?**\
> The distribution of synthetic samples used for mixup should not be too dissimilar to the train/test data. See the general response for a discussion of this. We measured FID values to quantify distance between distributions in Tables 2 and 3 (attached 1-page PDF). When the FID value is too large as is the case for EuroSAT, we observe little to no benefits from mixup with synthetic example, although there is still a benefit from "self-mixup" as evidenced by the significantly higher accuracy of Mixup-SelfAug compared to SelfAug. Table 5 in supplementary material (A.1) also discusses this but through a different lens.
> \
> \
> **6. Will it be possible to construct a mixup involving multiple samples (rather than self-mixup or diffusion model based) so that the gradients during training are more aligned, which thereby increases the signal-to-noise ratio?**\
> Ideally we would prefer to apply mixup to the original training samples directly as the reviewer suggests, i.e., mixup pairs of samples $(x, y)$ and $(x', y')$ where $(x, y)$ and $(x', y')$ are two randomly selected samples from the training set. As we explain in the paper (Sections 2.1 and 3.1) trying to apply this idea naively does not work. It doubles the sensitivity of the clipped-gradient sum in DP-SGD, which severely degrades the privacy guarantee (or requires doubling the scale of noise to achieve the same guarantee). The obvious way to get around this is to use microbatching as we explain in Section 3.1. But microbatching itself degrades performance because it has the effect of increasing noise [31]. So even though the gradients are more aligned the negative impact of noise results in worse models. We show this experimentally in Table 1 in the submission PDF, where we see that microbatching (of size 2) negatively impacts accuracy so much that it completely negates the benefits of mixup. Note: in that table ``baseline'' refers to vanilla DP-SGD. This is what motivated us to design Mixup-SelfAug and Mixup-Diffusion, which attempt to get the benefits of mixup without the drawback of microbatching (or worsening the DP guarantee).

---

### Author Rebuttal · Authors · 2023-08-10

Thank you for the feedback. We would like to clarify a few points.
\
\
**1. Motivation & novelty.**\
Data augmentation has the potential to improve DP-SGD, but naive application of techniques such as mixup compromises privacy. We propose two methods, Mixup-SelfAug and Mixup-Diffusion, to use mixup with DP-SGD *without* worsening privacy. Experimental results show that our methods consistently outperform the current SoTA (``Self-Aug'' from De et al. [10]) which uses single-sample data augmentation and serves as an important baseline.

Table 1 in the 1-page rebuttal PDF shows results on Caltech256, SUN397, Oxford-IIIT Pet. It shows that Mixup-SelfAug and Mixup-Diffusion outperform Self-Aug consistently across all datasets and privacy budgets tested. In some cases, the absolute test accuracy increase is particularly large (e.g., Caltech256 with $\varepsilon=1$).
\
\
**2. Comparison with De et al. [10].**\
De et al. [10] Self-Aug is the prior SoTA. They use single-sample data augmentation (SSDA) to improve performance over vanilla DP-SGD. Our proposed methods use a similar clipping idea to ensure DP. However, we show how to support multi-sample data augmentation (MSDA) such as mixup. Our experiments demonstrate that mixup provide substantial improvements over Self-Aug.
\
\
**3. Does more augmentations help?**\
Single-sample augmentations cannot provide benefits comparable to mixup for DP-SGD. For instance, Table 6 (attached 1-page PDF) shows that even with more augmentations, Self-Aug does not reach the performance of our proposed methods. In fact, additional augmentations (beyond $k=16 $ as recommended in [10]) decreases performance.
\
\
**4. What about different augmentations?**\
Adding augmentations beyond those suggested by De et al. [10] (which are flipping and cropping) hurts performance. We tested adding color jitter to the self-augmentation process of De et al. [10] (we call this Self-Aug+). (The idea of using color jitter is from Sander et al. We use their same parameter configuration. [https://github.com/facebookresearch/tan](https://github.com/facebookresearch/tan)) Results are shown in Table 4 (attached 1-page PDF), where we see that Self-Aug+ provides worse results than Self-Aug.
\
\
**5. Do our techniques benefit from additional augmentations?**\
By contrast Mixup-Diffusion benefits from additional diffusion samples $k'' $ as shown in Table 5 (attached 1-page PDF). For example, this leads to a 9.65% absolute test accuracy increase on Oxford-IIIT Pet.
\
\
**6. Selection of pre-training data, diffusion model and fair comparisons.**\
We took great care to ensure that our experiments lead to a fair comparison between our methods and alternatives such as Self-Aug (prior SoTA). In particular, all methods have access to the exact same training data. We also tune hyperparameters of each method optimally (e.g., $ k $ for Self-Aug). We use the same pre-trained models (Vit-B-16 and ConvNext from OpenClip) to compare our methods to others (Self-Aug and Mixed Ghost Clipping).

Since Mixup-Diffusion uses a diffusion model to generate synthetic examples, this could make the comparison unfair because other methods do not use synthetic samples. To avoid this, we purposefully use the exact same pre-training data (i.e., LAION-2B) to pre-train models as was used to train the diffusion model.  This avoids the issue of the synthetic examples somehow ``containing'' data that other methods do not have access to. Moreover, we conducted experiments (supplementary material A.2 --- Table 6) to show that the synthetic examples themselves do **not** boost performance. It is the **way** they are used by Mixup-Diffusion that boosts performance. Finally, out of the six datasets we use for evaluation, none of them overlap with the LAION-2B dataset (to the best of our knowledge).
\
\
**7. Explanations for Figure 2**\
Reviewers raised a valid point regarding the lack of clarity of Figure 2 in the submission PDF. The original was plotted using WRN-16-4 model trained from scratch on CIFAR-10 and did not include Mixup-Diffusion.

To address these concerns, we redid this experiment but in the pre-trained setting on CIFAR-10 and Caltech256 and including Mixup-Diffusion. See Figure 1 (attached 1-page PDF). To clarify: the figure plots the per-parameter gradient magnitude averaged over samples at each epoch. The histogram shows the data averaged over all training epochs and the X\% color lines show that data only for the epoch at X\% of the total training process. There are 10 epochs for this experiment, so for example the line for 20% shows the data for epoch 2 (out of 10).

The figure shows more concentrated values for our methods compared to the Self-Aug baseline, which suggests more stable training and faster convergence. Standard deviations for CIFAR-10 with Self-Aug, Mixup-SelfAug and Mixup-Diffusion are: $2.16 \cdot 10^{-3}$, $9.76 \cdot 10^{-4}$ and $9.59 \cdot 10^{-4}$, respectively. For Caltech256 they are: $1.43 \cdot 10^{-3}$, $1.07 \cdot 10^{-3}$ and $9.32 \cdot 10^{-4}$, respectively. This is consistent with experimental results of test accuracies for each method.
\
\
**8. Why do our techniques work?**\
We believe the data augmentation in general and mixup in particular helps generalization. Furthermore, if we view data augmentation as effectively increasing the amount of data available during training, then DP model training will particularly benefit from it. (This view is consistent with arguments from related work [42].)

To confirm our intuition, we perform two types of experiments. The first type quantifies the potential benefit from synthetic diffusion examples and is described in supplementary material (A.1 --- Table 5). The second type measures the Fréchet Inception Distance (FID) between train/test data and diffusion data for each dataset and shown in Tables 2 and 3 (attached 1-page PDF), where results are consistent with test accuracies across datasets and methods.

---

### Decision · Program_Chairs · 2023-09-21

**Decision:**

Accept (poster)

**Comment:**

This is a borderline paper.  Reviewer ayCk tends to reject this paper given the comments. During the rebuttal, this reviewer did not reply and provide further feedback.  AC read the rebuttal carefully and thought the concerns from the reviewer are valuable. However, AC tought the authors indeed did a good job to address them. AC hopes the authors can improve the paper based on all reviewers' comments.